# Oil-in-water emulsion adjuvants for pediatric influenza vaccines: a systematic review and meta-analysis

Yu-Ju Lin [1,2], Chiao-Ni Wen[1,3,4], Ying-Ying Lin[1,5], Wen-Chi Hsieh[1], Chia-Chen Chang[1], Yi-Hsuan Chen [1], Chian-Hui Hsu[1,5], Yun-Jui Shih[1,2], Chang-Hsun Chen[2] & Chi-Tai Fang [1,6]*

Standard inactivated influenza vaccines are poorly immunogenic in immunologically naive healthy young children, who are particularly vulnerable to complications from influenza. For them, there is an unmet need for better influenza vaccines. Oil-in-water emulsion-adjuvanted influenza vaccines are promising candidates, but clinical trials yielded inconsistent results. Here, we meta-analyze randomized controlled trials with efficacy data (3 trials, n = 15,310) and immunogenicity data (17 trials, n = 9062). Compared with non-adjuvanted counterparts, adjuvanted influenza vaccines provide a significantly better protection (weighted estimate for risk ratio of RT-PCR-confirmed influenza: 0.26) and are significantly more immunogenic (weighted estimates for seroprotection rate ratio: 4.6 to 7.9) in healthy immunologically naive young children. Nevertheless, in immunologically non-naive children, adjuvanted and non-adjuvanted vaccines provide similar protection and are similarly immunogenic. These results indicate that oil-in-water emulsion adjuvant improves the efficacy of inactivated influenza vaccines in healthy young children at the first-time seasonal influenza vaccination.

[1] Institute of Epidemiology and Preventive Medicine, College of Public Health, National Taiwan University, Taipei, Taiwan. [2] Taiwan Centers for Disease Control, Taipei, Taiwan. [3] Department of Laboratory Medicine, Linkou Chang Gung Memorial Hospital, Taoyuan, Taiwan. [4] Department of Medical Biotechnology and Laboratory Science, Chang Gung University, Taoyuan, Taiwan. [5] Center for Drug Evaluation, Taipei, Taiwan. [6] Division of Infectious Diseases, Department of Internal Medicine, National Taiwan University Hospital, Taipei, Taiwan. *email: fangct@ntu.edu.tw

Seasonal influenza caused 3.7 million illnesses, 25,644 hospitalizations, and 115 deaths among young children in the United State during 2017–2018 influenza season[1]. Inactivated influenza vaccines without adjuvant are poorly immunogenic in young children[2], who are at high risk for influenza-associated complications[3–5]. To improve the efficacy, a two-dose schedule (separated by a >4-week interval) is currently recommended for children younger than 9 years in the United States[6,7]. Nevertheless, the need for the second dose is cumbersome for parents[8,9]. Moreover, even after two-dose vaccination, the proportion of young children with a ≥1:40 hemagglutinin inhibition (HI) titer reaches only 70% for A/H1N1, 60% for A/H3N2, and <40% for B for children who are seronegative at baseline[2]. Therefore, there is an unmet need for better influenza vaccines.

Live-attenuated influenza vaccines showed a high efficacy against laboratory-confirmed influenza in children after a single dose in pre-marketing randomized trials[10,11]. NHS England recommended live-attenuated influenza vaccines for healthy young children aged >2 years since 2012[12,13]. Live-attenuated influenza vaccines were also recommended for children in Canada since the 2011/2012 season[14]. However, the US Centers for Disease Control and Prevention did not endorse live-attenuated influenza vaccines for 2016–2017 and 2017–2018 influenza seasons, because of their poor effectiveness against H1N1pdm09[7,15,16]. Although a new formulation of these vaccines was again approved for the 2018–2019 influenza season, an effectiveness estimate against H1N1pdm09 is still pending[16].

Another option is using adjuvants to enhance the immunogenicity of inactivated influenza vaccines. Aluminum salts, the traditional adjuvant in use for more than 80 years, have no satisfactory adjuvanticity for influenza vaccines[17,18]. In 1990s, it was discovered that oil-in-water emulsion is an effective adjuvant for influenza vaccine[19,20]. MF59 is an optimized oil-in-water formulation using biodegradable squalene and two nontoxic nonionic surfactants, Tween 80 and Span 85[21]. MF59-adjuvanted seasonal influenza vaccines were first licensed for the elderly in 1997[21]. AS03, another oil-in-water formulation of squalene, polysorbate 80, and α-tocopherol, allowed the mass production of pandemic influenza vaccines using reduced antigen dose during the 2009 global A(H1N1) pandemic[22].

Most studies of oil-in-water-adjuvanted influenza vaccines have been conducted in adults[23,24]. There are only limited and inconsistent data from children. Here, we conduct the first meta-analysis of data from clinical trials involving a total of more than 15,000 participants. We aim to evaluate whether oil-in-water-adjuvanted inactivated influenza vaccines provide protection superior to that by non-adjuvanted counterparts for children <9 years old, in terms of clinical efficacy against reverse transcription-polymerase chain reaction (RT-PCR)-confirmed influenza (any strain), seroprotection rate (the proportion with a ≥1:40 HI titer), and the need for a second dose. We also assess harm outcomes, including serious adverse event (SAE, defined as events that are life threatening, resulting in death, required hospitalization, resulting in a residual disability, or required intervention to prevent permanent impairment or damage), neurological events (defined as events that involved central or peripheral nervous system, including Guillain–Barré syndrome, cranial nerve palsy, narcolepsy, multiple sclerosis, and other neuroinflammatory disorders), and reactogenicity (fever or local pain/tenderness at injection sites).

## Results

**Results of the search.** Search strategies using keywords (Supplementary Notes 1–7) identified 2800 records, corresponding to 697 citations after removal of duplicates. We screened 697 records and excluded 626 records based on the title and abstract.

We retrieved the full text of 71 papers and excluded 53 papers (exclusion reasons are summarized in Supplementary Note 8). We included a total of 18 studies in the quantitative synthesis (Fig. 1).

All the 18 studies are randomized controlled trials (RCTs). Three trials reported efficacy data (Supplementary Table 1, these three trials also reported seroprotection rate data). Seventeen trials reported seroprotection rate data (Supplementary Table 2). Fifteen trials reported SAE, neurological events, and reactogenicity data (Supplementary Tables 3–5); 14 of these 15 trials also reported seroprotection rate data)[25–42]. Some trials consisted of several comparisons that assessed the same vaccines in different age groups or assessed different hemagglutinin antigen doses. Six of the 18 trials compared the same vaccine with or without adjuvants. The other trials compared adjuvanted and non-adjuvanted vaccines from different manufacturers (eight trials) or used antigen-sparing design with a reduced antigen dose in adjuvanted vaccines arm (five trials).

**Efficacy against RT-PCR-confirmed influenza.** Three clinical trials reported data on clinical efficacy against RT-PCR-confirmed influenza (five comparisons with a total of 15,310 participants). Vesikari et al.[39] (two comparisons) enrolled children with or without influenza vaccination history since 1 July 2010 (38–75% of their participants had a ≥1:40 HI titer to A/H1N1 or A/H3N2 at the baseline). The remaining two trials, Vesikari et al.[29] (two comparisons) and Nolan et al.[34] (one comparison) enrolled only vaccine-naive children.

We observed a very high between-trial heterogeneity (Fig. 2), with a total $I^2$ up to 84%. To investigate the source of heterogeneity, we divided the five comparisons trials into two subgroups: the all-naive group (the three comparisons that only enrolled vaccine-naive children, reported by Vesikari et al. in 2011[29] and Nolan et al. in 2014[34], respectively) and the non-naive group (the two comparisons involving non-naive children, up to 38–75%, reported by Vesikari et al. in 2018[39]).

In all-naive group, adjuvanted vaccines were associated with a significantly lower risk for RT-PCR-confirmed influenza than non-adjuvanted counterparts, with a weighted risk ratio estimate of 0.26 (95% confidence interval [CI]: 0.14–0.47; $I^2$: 0%). Vesikari et al.[29], which included a placebo arm, reported an absolute vaccine efficacy (against RT-PCR-confirmed influenza) of 86% (95% CI: 74–93%) for adjuvanted vaccines and 43% (95% CI: 15–61%) for non-adjuvant counterparts. Nolan et al.[34] did not include an all-placebo arm, and therefore did not report absolute vaccine efficacy.

On the other hand, in the non-naive group, there existed no significant difference in risk for RT-PCR-confirmed influenza between adjuvanted and non-adjuvanted arms (weighted risk ratio estimate: 0.92; 95% CI: 0.59–1.42; $I^2$: 80%).

**Seroprotection rate.** Of the 17 clinical trials with seroprotection rate data (26 comparisons with a total of 9062 participants), 12 to 21 of the 26 comparisons (six trials reported seroprotection against only H1N1[27,30–32,34,36], rather than against all three types of influenza virus) reported seroprotection rates 21–28 days after the first-dose vaccination, and 12 to 23 comparisons reported seroprotection rates 21–28 days after the second dose (Supplementary Table 2). Two trials (3 comparisons with a total of 157 participants) specifically enrolled primed children (who had received prior influenza vaccination)[26,37].

The comparators were non-adjuvanted trivalent vaccines in 16 of the 17 trials. Only one trial (with 1788 participants) compared adjuvanted versus non-adjuvanted quadrivalent influenza vaccine[39]. For this trial, we included data on seroprotection rate

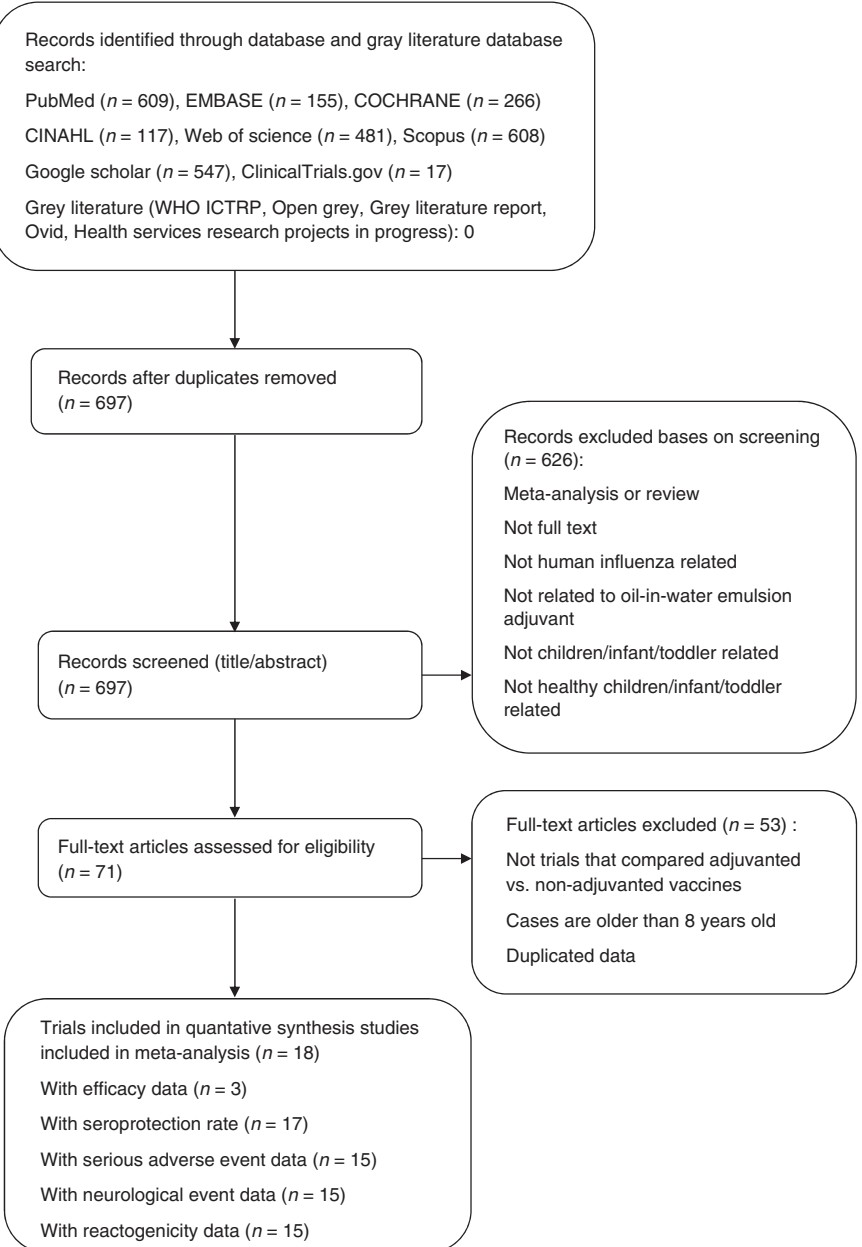

**Fig. 1** Flowchart of the literature search.

against H1N1, H3N2, and B/Yamagata in the meta-analysis based on recommendation by the World Health Organization for trivalent vaccines in the study year.

We observed a very high heterogeneity between trials in the adjuvanticity effect on seroprotection against A/H1N1 (Fig. 3), with a total $I^2$ of 99%. To investigate the source of heterogeneity, we performed meta-regression with seroprotection rate ratio (RR) (between adjuvanted and non-adjuvanted arms) as the dependent variable and the independent variable was the inverse of seroprotection rate in the non-adjuvanted arm (a surrogate indicator for whether the young children partici-pants were immunologically naive to influenza or influenza vaccines). Figure 4 showed that heterogeneity in RR after the first dose was nearly all explained by serological response rate in non-adjuvanted arm, with an $R^2$ of 99.79%. Meta-regression of RR after the second dose (Supplementary Fig. 1) yielded the same findings (Supplementary Fig. 2).

We further divided all studies into three subgroups, based on the seroprotection rate after a single dose of non-adjuvanted vaccine: poor response (<25%), moderate response (25–75%), and good response (>75%). Figure 3 showed that, in poor response group, seroprotection after adjuvanted influenza vaccines were significantly superior to that after non-adjuvanted counterparts with a weighted estimate for RR of 4.6 (95% CI: 4.1–5.2; $I^2$: 0%). In contrast, in good response group, both arms had equally good post-vaccination seroprotection rate (weighted RR: 1.0; 95% CI: 0.96–1.04; $I^2$: 0%).

High heterogeneity also exists in adjuvanticity effect on seroprotection against A/H3N2 (total $I^2$ 99%) (Fig. 5 and Supplementary Fig. 3). Meta-regression shows that between-trials heterogeneity in RR was nearly all explained by serological response rates in the non-adjuvanted arm, with an $R^2$ of 99.31% (Fig. 6 and Supplementary Fig. 4). Figure 5 shows that, in poor response group, seroprotection after adjuvanted vaccines were

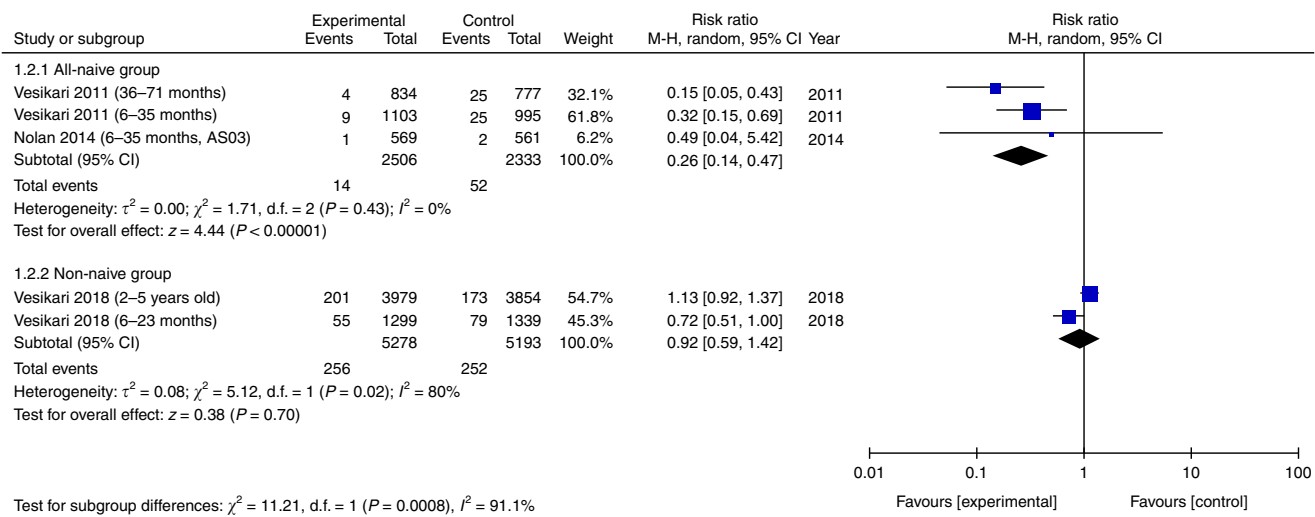

**Fig. 2** Forest plot showing the risk ratios of RT-PCR-confirmed influenza among children who received adjuvanted versus non-adjuvanted trivalent vaccines or quadrivalent vaccines.

**Fig. 3** Forest plot showing the ratios of the seroprotection rate (SPR) against H1N1, defined as a ≥1:40 hemagglutinin inhibition (HI) titer, 21–28 days after the first dose among children who received adjuvanted versus non-adjuvanted trivalent vaccines (TIVs) or quadrivalent vaccines (QIVs).

significantly superior to that after non-adjuvanted counterparts (weighted RR: 7.9; 95% CI: 6.7–9.4; $I^2$: 0%). In contrast, in the good response group, adjuvanted and non-adjuvanted arms had an equally good post-vaccination seroprotection rate (weighted RR: 1.0; 95% CI: 0.97–1.05; $I^2$: 0%).

For seroprotection against influenza B, adjuvanticity was less impressive after the first dose in poor response group (Supplementary Fig. 5). Only after the second dose, adjuvanted vaccines were significantly better than non-adjuvanted counterparts in poor response group (weighted RR: 5.4; 95% CI: 3.8–7.5; $I^2$: 24%)

(Fig. 7). Heterogeneity was also high (total $I^2$ 59%). Meta-regression shows that between-trials heterogeneity was nearly all explained by the serological response rates in the non-adjuvanted arm ($R^2 = 99.6\%$) (Fig. 8). Similarly, in the good response group, there was a lack of significant difference between adjuvanted and non-adjuvanted arms (weighted RR: 1.1; 95% CI: 1.0–1.2; $I^2$: 65%).

A pooled analysis of all the included trials showed that after a single dose of oil-in-water-adjuvanted influenza vaccine, the proportion of children with a ≥1:40 HI titer reached 88.7% (3419/3853) for A/H1N1 and 87.7% (2803/3196) for A/H3N2, but only 36.2% (1158/3196) for B. Only after the second dose, proportions

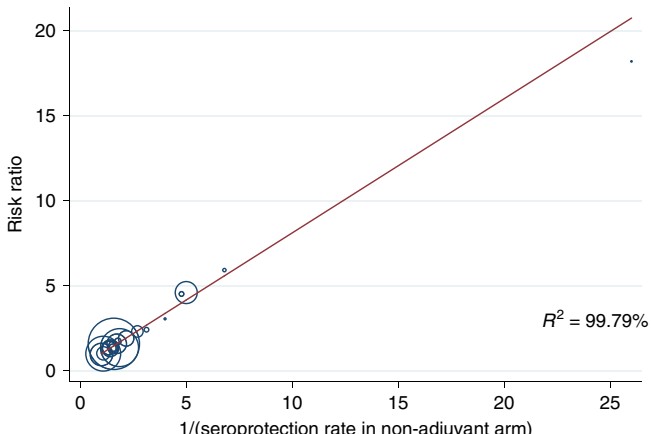

**Fig. 4** Meta-regression showing a linear relationship (slope = 0.79, $P < 0.001$) between the ratios of the seroprotection rate against H1N1 after the first dose and the inverse of seroprotection rate in non-adjuvanted arm, with an $R^2$ of 99.79%.

of young children with a ≥1:40 HI titer against influenza B virus reach 89.0% (3225/3623). In contrast, after two doses of non-adjuvanted influenza vaccines, the proportions of children with a ≥1:40 HI titer reached only 71.8% (3141/4377) for A/H1N1, 73.9% (2621/3544) for A/H3N2, and 48.9% (1687/3451) for B (Supplementary Figs. 1, 3, and 5).

**Serious adverse event**. Fifteen trials reported a total of 1138 SAEs, which may or may not be considered vaccine-related by trial investigators, from 25,541 participants during follow-up period (ranging from 21 to 390 days after vaccination, Supplementary Table 3). The most common SAE were gastroenteritis, pneumonia, convulsion, febrile convulsion, appendicitis, and asthma.

Compared with participants in non-adjuvanted arms, those in adjuvanted arms did not have a higher frequency of SAE (weighted SAE risk ratio: 0.9; 95% CI: 0.7–1.1; $I^2$: 50%; Fig. 9) during the follow-up period.

**Neurological events**. Fifteen trials reported a total of 13 neurological events (nine in adjuvanted arm and four in non-adjuvanted arm) from 20,802 participants during follow-up period (ranging from 21 to 390 days after the vaccination, Supplementary Table 4). These neurological events include febrile convulsion (7/13), convulsion (3/13), and multiple seizure (3/13). No narcolepsy cases were reported.

Compared with participants in non-adjuvanted arms, those in adjuvanted arms had a slightly higher frequency of neurological events during the follow-up period, but the difference did not reach statistical significance due to the very low frequency of events (weighted risk ratio estimate: 1.4; 95% CI: 0.4–4.3; $I^2$: 0%; Fig. 10).

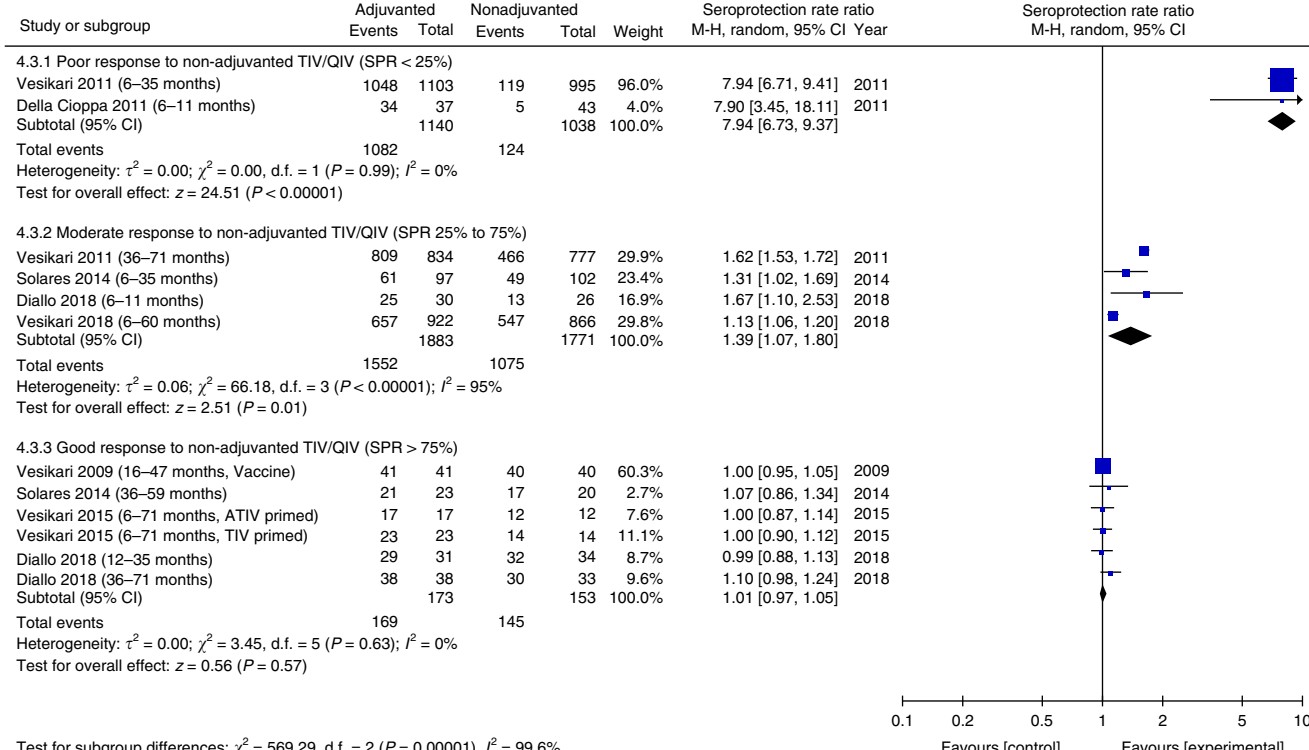

**Fig. 5** Forest plot showing the ratios of the seroprotection rate (SPR) against H3N2, defined as a ≥1:40 hemagglutinin inhibition (HI) titer, 21–28 days after the first dose among children who received adjuvanted versus non-adjuvanted trivalent vaccines (TIVs) or quadrivalent vaccines (QIVs).

**Reactogenicity**. Of the 15 trials (21 comparisons) with reactogenicity data, all the 21 comparisons (some comparisons reported only pain or fever, rather than fever or local pain at the injection sites) reported reactogenicity 7 days after the first-dose vaccination, and 18 comparisons reported reactogenicity 7 days after the second dose (Supplementary Table 5).

Compared to non-adjuvanted vaccines, oil-in-water-adjuvanted vaccines were associated with a weighted estimate of a 1.3-fold increase in pain/tenderness at the injection site (95% CI: 1.2–1.5; Supplementary Fig. 6) after the first dose. The weighted estimate

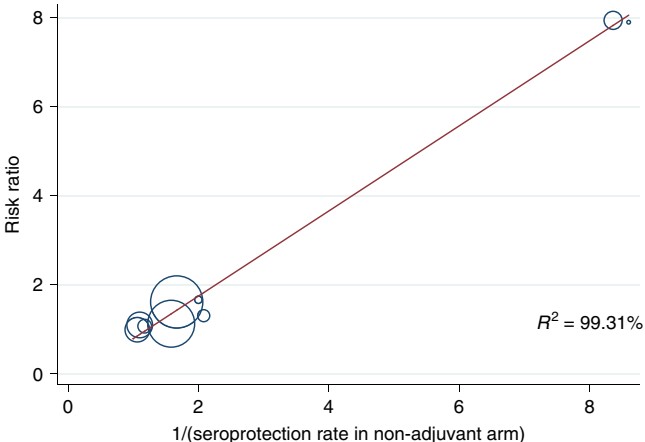

**Fig. 6** Meta-regression showing a linear relationship (slope = 0.96, $P <$ 0.001) between the ratios of the seroprotection rate against H3N2 after the first dose and the inverse of seroprotection rate in non-adjuvanted arm, with an $R^2$ of 99.31%.

for fever after the first dose was 1.6-fold (95% CI: 1.4–1.8; Supplementary Fig. 7). After the second dose, there was a 1.5-fold increase in pain/tenderness at the injection site (95% CI: 1.2–1.8; Supplementary Fig. 8) and a 1.5-fold increase in risk of fever (95% CI: 1.2–1.9; Supplementary Fig. 9).

Funnel plot analyses did not show publication biases (Supplementary Fig. 10).

**Risk of bias assessment**. Two trials, Langley et al.[31] and Zedda et al.[38], had a high risk of bias. Langley et al.[31] was observer blind and had a small sample size in one comparison (five versus four participants). Zedda et al.[38] did not report whether allocation concealment had been used. The other trials had a low-to-moderate risk of bias. The risk of bias assessment results are summarized in Supplementary Figs. 11 and 12.

**Sensitivity analysis**. Exclusion of two trials with high risk of bias[31,38] did not change the weighted estimates for seroprotection RR or meta-regression results (Supplementary Figs. 13–18). The $R^2$ of meta-regression model remains very high (99.81% for A/H1N1, 88.54% for A/H3N2, and 99.63% for B).

Exclusion of the 12 trials that did not compare the same vaccine (with or without adjuvants) did not significantly alter the weighted estimate for risk ratio of RT-PCR-confirmed influenza in all-naive group (0.2; 95% CI: 0.1–0.5), the weighted estimate for seroprotection RR in the poor response group (H1N1: 4.6, 95% CI: 4.1–5.2; H3N2: 7.9, 95% CI: 6.7–9.4; B: 5.0, 95% CI: 4.4–5.7), the meta-regression results ($R^2$ 99.56% for H1N1, 100% for H3N2, 99.84% for B), and the weighted estimate for SAE risk ratio (0.7; 95% CI: 0.5–0.8), as well as the weighted estimate for

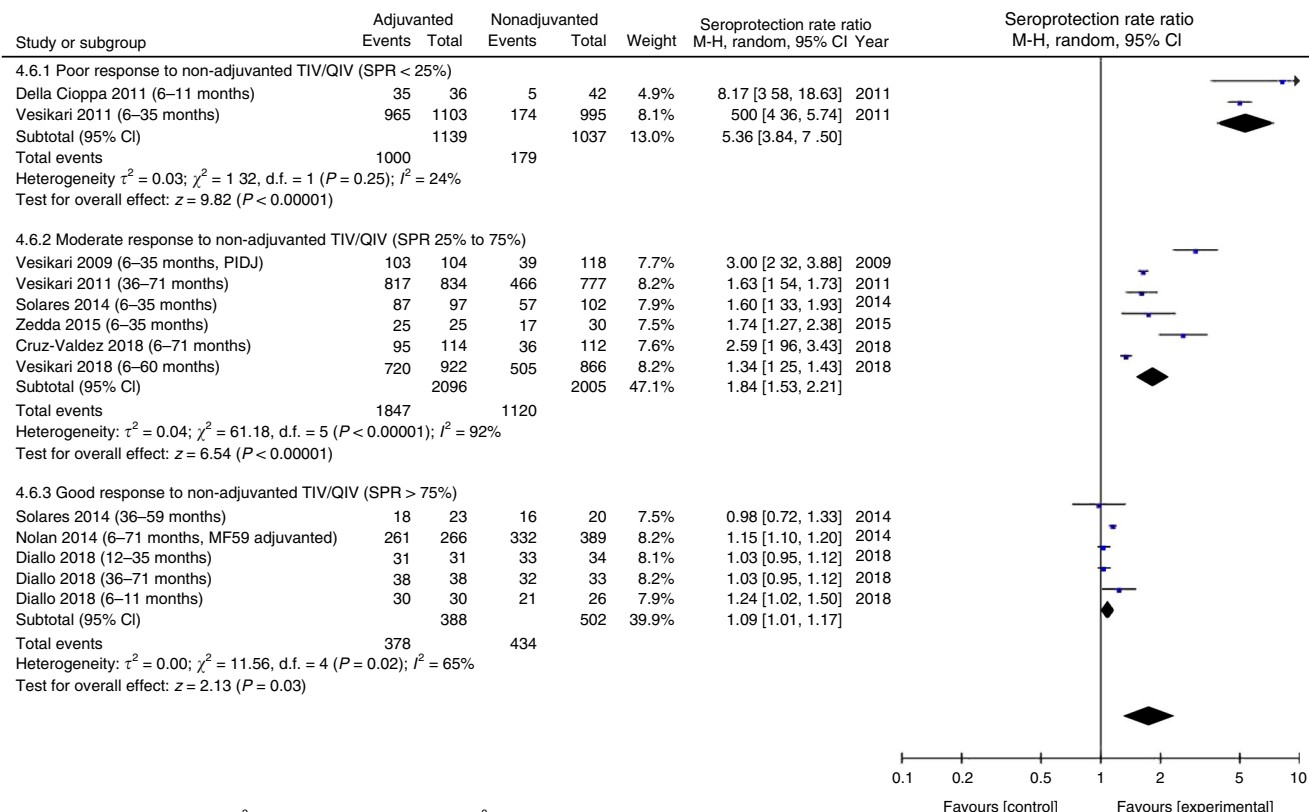

**Fig. 7** Forest plot showing the ratios of the seroprotection rate (SPR) against B, defined as a ≥1:40 hemagglutinin inhibition (HI) titer, 21–28 days after the second dose among children who received adjuvanted versus non-adjuvanted trivalent vaccines (TIVs) or quadrivalent vaccines (QIVs).

risk ratio of neurological events (2.8; 95% CI: 0.1–67.5) (Supplementary Figs. 19–27).

To examine whether the near perfection correlation in meta-regression is driven by a few outlying data points, we conducted a sensitivity analysis without these potential outliers. After excluding those data points with an inverse of seroprotection rate in non-adjuvanted arm larger than 5, the $R^2$ of meta-regression model remains very high ($R^2$ 99.67% for A/H1N1, and 96.21% for B) except that for A/H3N2 ($R^2$ 42.98%), in which the meta-regression results became imprecise (slope = 0.47; $P = 0.075$) when the number of data points decreased to only seven (Supplementary Figs. 28–30).

**Quality of evidence**. We used the GRADE approach to assess the quality of evidence, taking risk of bias, risk of random errors, risk of publication bias, and risk of lack of external validity into consideration. We summarize the findings in Supplementary Table 6 (effect outcomes) and Supplementary Table 7 (harm outcomes).

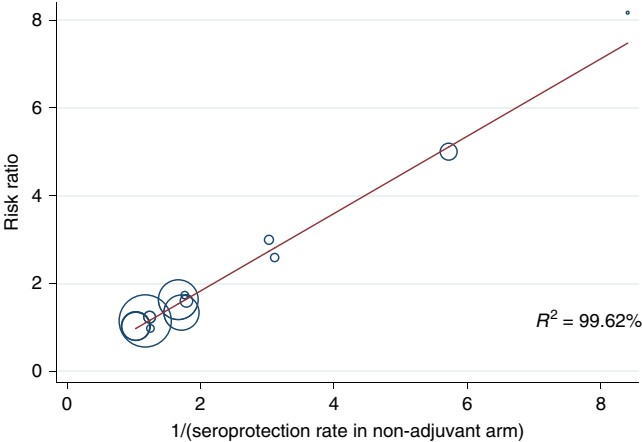

**Fig. 8** Meta-regression showing a linear relationship (slope = 0.88, $P < 0.001$) between the ratios of the seroprotection rate against B after the second dose and the inverse of seroprotection rate in non-adjuvanted arm, with an $R^2$ of 99.62%.

The certainty of evidence was high for the RT-PCR-confirmed influenza endpoints due to large effect (risk ratio <0.5) based on consistent evidence from two RCTs. However, there were potential outcome reporting bias (only 3 of the 18 included trials reported RT-PCR-confirmed influenza endpoints).

The certainty of evidence was moderate for seroprotection rate endpoints (against H1N1 after the first dose, against H3N2 after the first dose, and against B after the second dose) due to potential outcome reporting bias (only 12 to 21 of the 26 included comparisons reported seroprotection rate data after the first vaccine dose) and indirectness (seroprotection was used as a surrogate for real-life protection against infection, disease and death), despite large effect (risk ratio >2.0) based on consistent evidence from at least two RCTs.

The certainty of evidence was very low for SAE endpoint due to potential outcome reporting bias (only 15 of the 18 included trials reported SAE data), heterogeneity between trials ($I^2$ 50%), and wide CI.

The certainty of evidence was low for neurological event endpoint due to potential outcome reporting bias (only 15 of the 18 included trials reported neurological events data), and few events with a wide CI.

The certainty of evidence of pain at the injection site after the first and second doses, as well as of fever after the second dose, was low due to potential outcome reporting bias (only 17 to 20 of the 21 included comparisons reported pain/fever rate data), and heterogeneity between comparisons ($I^2$: 46–81%).

## Discussion

This is the first systematic review and meta-analysis of the efficacy and harm of oil-in-water adjuvants for seasonal influenza vaccines in young children. Compared with non-adjuvanted counterparts, adjuvanted influenza vaccines provided a significantly better protection and were significantly more immunogenic in healthy immunologically naive young children. However, in immunologically non-naive children, adjuvanted and non-adjuvanted vaccines provided similar protection and were similarly immunogenic. Participants in adjuvanted vaccine arms had a small increase in frequency of neurological events, which should be considered as a potential hazard that would require close and specialized monitoring regardless of statistical significance.

| Study or subgroup | Experimental Events | Total | Control Events | Total | Weight | Risk ratio M-H, random, 95% CI |
|---|---|---|---|---|---|---|
| Arguedas 2011 | 0 | 56 | 1 | 54 | 0.6% | 0.32 [0.01, 7.73] |
| Block 2012 | 6 | 161 | 16 | 161 | 5.5% | 0.38 [0.15, 0.93] |
| Cruz-Valdez 2018 | 0 | 140 | 0 | 137 | | Not estimable |
| Della Cioppa 2011 | 4 | 37 | 1 | 43 | 1.2% | 4.65 [0.54, 39.78] |
| Diallo 2018 | 0 | 118 | 0 | 119 | | Not estimable |
| Langley 2012 | 4 | 59 | 0 | 57 | 0.7% | 8.70 [0.48, 157.01] |
| Nassim 2012 | 6 | 334 | 6 | 334 | 3.9% | 1.00 [0.33, 3.07] |
| Nolan 2014 (AS03) | 76 | 2048 | 68 | 2049 | 18.6% | 1.12 [0.81, 1.54] |
| Nolan 2014 (MF59) | 125 | 3123 | 59 | 1474 | 19.4% | 1.00 [0.74, 1.35] |
| Solares 2014 | 0 | 180 | 2 | 180 | 0.6% | 0.20 [0.01, 4.14] |
| Vesikari 2009 (PIDJ) | 2 | 130 | 6 | 139 | 2.1% | 0.36 [0.07, 1.73] |
| Vesikari 2009 (Vaccine) | 0 | 43 | 0 | 46 | | Not estimable |
| Vesikari 2011 | 122 | 2012 | 169 | 1846 | 22.5% | 0.66 [0.53, 0.83] |
| Vesikari 2015 | 1 | 33 | 0 | 24 | 0.6% | 2.21 [0.09, 51.93] |
| Vesikari 2018 | 234 | 5243 | 230 | 5161 | 24.3% | 1.00 [0.84, 1.20] |
| | | | | | | |
| Total (95% CI) | | 13717 | | 11824 | 100.0% | 0.88 [0.69, 1.12] |
| Total events | 580 | | 558 | | | |

Heterogeneity: $\tau^2 = 0.05$; $\chi^2 = 21.93$, d.f. = 11 ($P = 0.02$); $I^2 = 50\%$
Test for overall effect: $z = 1.04$ ($P = 0.30$)

Risk ratio M-H, random, 95% CI — Favours [experimental] / Favours [control] (0.01, 0.1, 1, 10, 100)

**Fig. 9** Forest plot showing the risk ratios of serious adverse events during the follow-up time (ranging from 21 to 390 days) after the vaccinations among children who received adjuvanted versus non-adjuvanted trivalent vaccines (TIVs) or quadrivalent vaccines (QIVs).

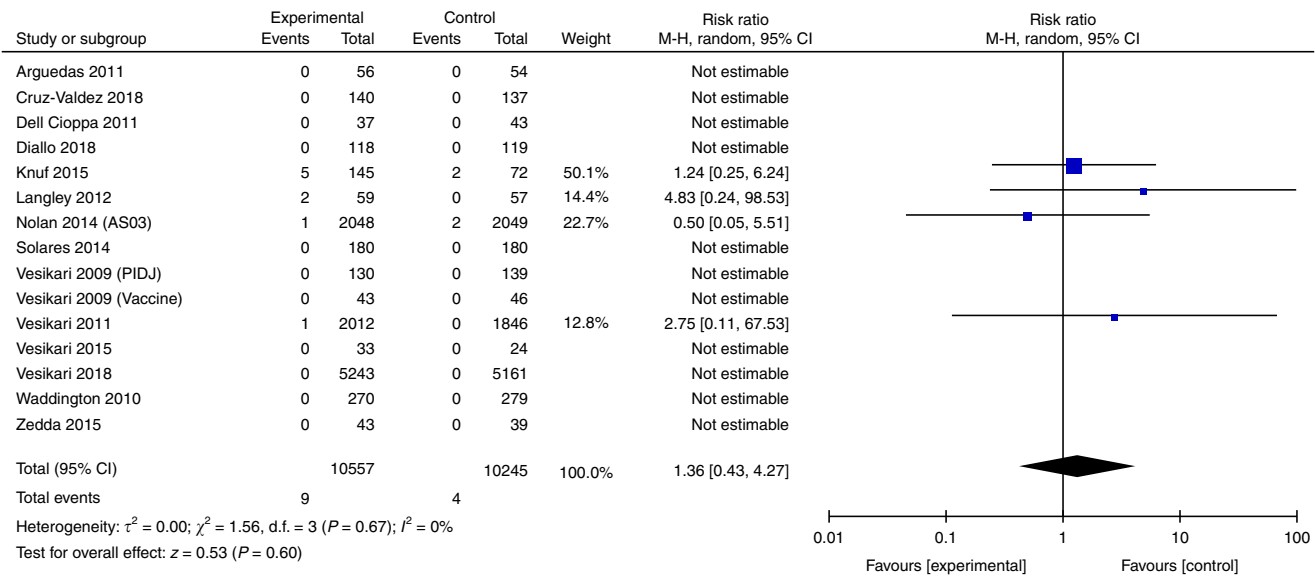

**Fig. 10** Forest plot showing the risk ratios of neurological events during the follow-up time (ranging from 21 to 390 days) after the vaccinations among children who received adjuvanted versus non-adjuvanted trivalent vaccines (TIVs) or quadrivalent vaccines (QIVs).

Mouse experiments showed that MF59 stimulates the local inflammatory response at the injection site and thus increases the uptake of antigen by activated dendritic cells[43]. Similarly, AS03 facilities a stronger antibody response by inducing a local cytokine and chemokine response at the injection site to enhance antigen uptake[43]. Studies in young children also showed that MF59-adjuvanted influenza vaccines produced a stronger and more homogeneous innate transcription response like that of adults, thus resulting in significantly higher HI titers than those elicited by non-adjuvanted vaccines[22].

The clinical endpoint is the gold standard in assessing vaccine efficacy[43]. To date, three randomized trial assessed the comparative efficacy of MF59-adjuvanted influenza vaccines against RT-PCR-confirmed influenza in young children. The first large trial (Vesikari et al.[29], with >1000 participants) reported that the MF59-adjuvanted influenza vaccine conferred protection that was significantly superior to that of non-adjuvanted counterparts (absolute vaccine efficacy of 86% versus 43%)[29]. However, good clinical practice (GCP) inspection found that "the kit for the extraction of viral nucleic acid in swab samples was not validated," and that the RT-PCR technique used to detect influenza virus "did not incorporate an internal control."[44] The inspection also identified "critical issues affecting the reliability of recorded adverse event and suspected influenza cases."[45] The authors "re-tested all available original samples (1208 of 1216 samples) that were collected during the trial in a qualified laboratory using a validated assay."[45] Although "analyses of the re-test results were remarkably similar to the original analyses, confirming the original overall study conclusions regarding the high absolute and relative efficacy,"[44] this MF59-adjuvanted pediatric influenza vaccine (Fluad Pediatric[TM]) was subsequently withdrawn from Europe in 2012 due to non-compliance with GCP during this clinical trial[44].

In addition to the above-mentioned GCP issues during the clinical trial, an important reason behind the decision to withdraw Fluad Pediatric[TM] from the market was a major concern raised by the inconsistency of the seroprotection data during the 2012 assessment[45]. In some of the trials, Fluad Pediatric[TM] did not yield a better seroprotection rate than its non-adjuvant counterparts[25,45]. This inconsistency further weakened the

quality of evidence supporting the use of oil-in-water emulsion adjuvants in children.

Our systematic review confirmed the very high heterogeneity between randomized trials assessing oil-in-water-adjuvanted influenza vaccines. To identify the source of heterogeneity, we performed meta-regression, which showed that this high between-trial heterogeneity was nearly all explained by the large variations in seroprotection rate in the non-adjuvanted arm (ranging from 4 to 100% across the trials). A high seroprotection rate in non-adjuvanted vaccines (probably the result of a high proportion of participants with a priming effect from previous exposure to natural influenza infections or influenza vaccination) poses a constraint on the numerical value of seroprotection RR. For example, with a 100% rate in the non-adjuvanted arm, the maximal possible RR between adjuvanted and non-adjuvanted groups would be only 1.0 (100% versus 100%). However, with a 4% rate in the non-adjuvanted arm, the maximal possible numerical value of RR would increase to 25 (100% versus 4%). By focusing on the trials with participants who were influenza vaccine naive and showing a poor serological response to non-adjuvanted vaccines, our meta-analysis was able to show that oil-in-water emulsion-adjuvanted vaccines consistently had a significantly superior efficacy and a significantly superior seroprotection rate, especially for A/H1N1 and A/H3N2.

Our findings support a role of oil-in-water-adjuvanted inactivated influenza vaccines for influenza-naive healthy young children, but not for children who had past influenza infections. Because flu-like symptoms in young children may be caused by other virus such as respiratory syncytial virus, an age-based approach would be desirable. The all-naive group (Fig. 2) and the poor response group (Figs. 3, 5, and 7), for whom adjuvanted vaccines are significantly superior to the non-adjuvanted counterparts, are mostly comprised of young children aged <3 years, while the non-naive group and the moderate/good response groups (for whom adjuvant and non-adjuvanted vaccines are similar) included many children aged 3 years or more. Therefore, oil-in-water-adjuvanted influenza vaccine may be most appropriate for the first seasonal influenza vaccination for young children <3 years old. On the other hand, for older children or children who had prior influenza vaccination, there might be no

advantage to using oil-in-water-adjuvanted influenza vaccines to replace less expensive non-adjuvanted vaccines for annual influenza vaccination.

Traditionally, the European Committee for Medicinal Products for Human Use (CHMP) used a set of serological criteria to evaluate influenza vaccines for adults[43]. The CHMP criteria were based on correlations between HI titer and infection rate observed in challenge studies conducted in healthy adults using attenuated influenza virus[43]. For seasonal influenza vaccines, a >2.5-fold increase in the geometric mean titer, a >40% seroconversion rate/ significant increase, or a >70% seroprotection rate was required for licensing[43]. The applicability of these criteria to young children has not been established[43]. Based on these criteria and given their recognized limitations, our meta-analysis showed that a single-dose oil-in-water emulsion-adjuvanted influenza vaccine might suffice for providing protection against A/H1N1 (pooled data for seroprotection rate: 88.7%) and A/H3N2 (pooled data for seroprotection rate: 87.7%). However, to provide protection against B, the second dose would be necessary because the seroprotection rate reached only 36.2% after the first dose.

The new CHMP regulatory guidelines, published in 2014, no longer use the traditional serological criteria[43,46]. For young children aged 6–36 months, the new guidelines require that randomized trials with clinical endpoints be conducted[43]. The high heterogeneity among previous randomized trials on oil-in-water-adjuvanted vaccines observed in our meta-analysis, which might be at least partly explained by the priming effect from natural influenza infection, highlights the necessity of taking the serostatus of participants into account when designing future randomized trials for these products[47].

In 2010, an increase in narcolepsy cases among children/ adolescents vaccinated with Pandemrix[TM] (AS03-adjuvanted H1N1pdm09-monovalent vaccine made by GlaxoSmithKline) was detected in Europe, with an absolute risk of one additional case per 12,000 to 27,800 vaccinated[48–50]. Narcolepsy incidences following other adjuvanted H1N1pdm09-monovalent vaccines were also investigated in Canada (where a different AS03-adjuvanted vaccine, Arepanrix[TM] from GlaxoSmithKline, was used)[51,52], South Korea (where MF59-adjuvanted vaccine Greenflu-S plus[TM] was used; the MF59 was from Novartis)[53], and Taiwan (where MF59-adjuvanted vaccine Focetria[TM] from Novartis was used)[54]. Most of these investigations did not find an increase in narcolepsy incidence, except for that in Quebec, Canada, where a possible small increase (one per million vaccine doses) was reported (although "a confounding effect of influenza infection cannot be ruled out"[52]). In Taiwan, a three-fold increase in narcolepsy incidence occurred before the start of H1N1 vaccination, and influenza-like illness during 2009 H1N1 pandemic period was a strong risk factor (case–control matched odds ratio: 2.72; $P = 0.0137$, by conditional logistic regression) for the onset of narcolepsy in children[54]. In China (where no adjuvanted H1N1pdm09-monovalent vaccines were used), narcolepsy incidence increased by 3-fold following the 2009 H1N1 pandemic, but the majority (>90%) of Chinese narcolepsy cases had no influenza vaccination history[55]. Antibodies to nucleoproteins of inactivated influenza virus in Pandemrix[TM], which cross-react with human hypocretin receptor 2, may trigger the immune-mediated pathogenesis of narcolepsy[56]. The other two AS03- or MF59-adjuvanted H1N1 vaccines, Arepanrix[TM] and Focetria[TM], contain a much lower amount of viral nucleoproteins than that in Pandemrix[56,57], and were not associated with a detectable increase in risk of narcolepsy[48–50,56–59]. However, it remains unknown whether removal of viral nucleoprotein from adjuvanted influenza vaccines, through a better purification process during manufacturing, would eliminate the risk of narcolepsy.

No narcolepsy case was reported in the RCTs, which were included in this systematic review and meta-analysis. The 13 reported neurological events were febrile convulsion or seizure. Given that narcolepsy is a rare adverse event following vaccination and the fact that there are frequently delays in making the diagnosis, this outcome could not be realistically examined in included trials because of the limited sample size (20,802 participants in total) and the relatively short follow-up duration (from 21 to 390 days). Therefore, the finding of an increased frequency of neurological events is concerning. This highlights the need for a very good vaccine safety system to tracking such outcome for adjuvanted pediatric vaccines in future clinical trials.

An important limitation of our study is the generalizability. Mismatch between vaccine strains and the circulating influenza strains may substantially decrease both the absolute vaccine efficacy and the relative advantage of adjuvanted vaccines seen in trials, as well as the statistical power to detect differences. Therefore, pre-licensing RCTs need to be followed by post-marketing field evaluation to quantify the real-world experience of vaccine effectiveness. The present systematic review was essentially based on data from highly controlled conditions of RCTs. We were unable to find any field evaluation reports, likely due to the withdrawal of Fluad Pediatric[TM] in 2012. Another important limitation is the limited length of follow-up (only up to 390 days after vaccination). Thus, long-term safety of these adjuvanted vaccines remains unknown.

In conclusion, our meta-analysis shows that oil-in-water emulsion adjuvants have good efficacy in enhancing both seroprotection rate and protection against laboratory-confirmed influenza for immunologically naive healthy young children receiving seasonal influenza vaccines for the first time. This benefit must be weighed against a possible small increase in risk of neurological events.

## Methods

**PROSPERO registration.** The study protocol was registered at PROSPERO (https://www.crd.york.ac.uk/prospero/) as CRD42018093436. We strengthened search strategy and performed meta-regression analysis, based on AMSTAR2 checklist[60], as recommended during the review process.

**Types of studies.** We considered RCTs, quasi-RCTs, comparative controlled trials, cohort studies, and case–control studies.

**Types of participants.** Healthy children under 9 years of age in any geographical location. All participants were classified as healthy unless otherwise stated. We excluded studies that documented the inclusion of participants with specific chronic pathology (i.e., diabetes or cardiac disease) or immunodeficiency, and studies assessing vaccine efficacy in selected groups affected by a specific chronic illnesses/conditions or immunodeficiency.

**Types of interventions.** Vaccination with any oil-in-water-adjuvanted influenza vaccines given independently, in any dose, preparation, or time schedule (intervention), compared with non-adjuvanted influenza vaccines (reference).

**Definition of seroprotection.** Seroprotection is defined as a titer of ≥1:40 (or ≥1:32) by HI assay, as pre-specified by the investigators of each trial report.

**Definition of SAE.** SAE is defined as any post-vaccination event which is life threatening, resulting in death, required hospitalization, resulting in a residual disability, or required intervention to prevent permanent impairment or damage. SAE may or may not be vaccine related.

**Definition of neurological event.** Neurological event was defined as any post-vaccination event involving central or peripheral nervous system, including Guillain–Barré syndrome, cranial nerve palsy, narcolepsy, multiple sclerosis, and other neuroinflammatory disorders.

**Main outcome.** The main effect outcomes are (1) the ratio of RT-PCR-confirmed influenza risk (any strain) during the period 6–12 months after receiving

adjuvanted seasonal influenza vaccines versus that of non-adjuvanted counterparts; and (2) the ratio of the seroprotection rate (the proportion with a ≥1:40 HI titer) 21–28 days after receiving the first dose of adjuvanted seasonal influenza vaccines versus that of non-adjuvanted counterparts.

The main harm outcomes are the risk ratios of SAE and neurological events after receiving adjuvanted seasonal influenza vaccines versus that of non-adjuvanted counterparts, up to the end of follow-up.

**Secondary outcomes.** We assessed other potentially important outcomes, including the ratio of the seroprotection rate 21–28 days after receiving the second dose of adjuvanted seasonal influenza vaccines versus that of non-adjuvanted counterparts; and the risk ratios of (a) pain/tenderness at the injection site during the 7 days after the first dose; (b) pain/tenderness at the injection site during the 7 days after the second dose; (c) fever (body temperature higher than 38 °C) during the 7 days after the first dose; and (d) fever (body temperature higher than 38 °C) during the 7 days after the second dose.

**Electronic searches.** We searched the following bibliographic databases: ClinicalTrials.gov (www.clinicaltrials.gov, on 30 September 2019), Cochrane Central Register of Controlled Trials (CENTRAL; 2019, Issue 2 of 12, on 30 September 2019 via the Cochrane Library), Cumulative Index to Nursing and Allied Health Literature (CINAHL) (EBSCOhost) (1982 to 30 September 2019), Embase (Elsevier) (1974 to 30 September 2019), Google Scholar (as to 30 September 2019), MEDLINE (PubMed) (1950 to 30 September 2019), Scopus (1996 to 30 September 2019), Web of Science (Clarivate Analytics) (1900 to 30 September 2019), and World Health Organization International Clinical Trials Registry Platform (WHO ICTRP) (www.who.int/ictrp/en, 30 September 2019).

We also searched the following grey literature databases: Agency for Healthcare Research and Quality (AHRQ) Grants On-Line Database (https://gold.ahrq.gov/projectsearch/), Grey Literature Report (The New York Academy of Medicine) (http://www.greylit.org/faqhelp, data updated to 2016 only), Health Services and Sciences Research Resources (HSRR) (https://hsrr.nlm.nih.gov/?cf_redirect=t), HSRProj (Health Services Research Projects in Progress) (https://wwwcf.nlm.nih.gov/hsr_project/home_proj.cfm), Open Grey (http://www.opengrey.eu/), and Research Portfolio Online Reporting Tools (RePORT) (https://projectreporter.nih.gov/reporter.cfm).

We used the search strategy in Appendix Supplementary Note 1 to search MEDLINE (PubMed). We adapted the search terms to search the CENTRAL (Supplementary Note 2), Embase (Supplementary Note 3), CINAHL/Web of Science/Scopus (Supplementary Note 4), Google Scholar (Supplementary Note 5), WHO ICTRP (Supplementary Note 6), and ClinicalTrials.gov (Supplementary Note 7). We searched literature in all relevant languages.

**Selection of studies.** Screening was done in duplicate. Two review authors (Y.-J.L., C.-N.W.) independently applied inclusion criteria to all identified and retrieved literature. Any difference on which study to include was resolved by consensus. We listed all excluded studies in Supplementary Note 8, along with the reasons of exclusion.

**Data extraction and management.** Extraction was done in duplicate. Two review authors (Y.-J.L., C.N.W.) independently performed data extraction using a data extraction form. Any difference was resolved by consensus. Y.-J.L and C.N.W. entered the data into Review Manager 5.3 (RevMan 2014). We extracted data on the following: Study design and quality, Demographic data of participants, Vaccine strains, Main outcomes and secondary outcomes, Follow-up duration, and Funding sources.

**Data synthesis and heterogeneity assessment.** The weighted RT-PCR-confirmed influenza risk ratio, seroprotection RR, and adverse events risk ratio as well as the 95% CIs were estimated by Mantel–Haenszel method and summarized by Forest plots. We used funnel plot to detect publication bias. Heterogeneity between studies was assessed by $I^2$ statistic, which describes the percentage of variation across studies that is due to heterogeneity rather than chance and is independent on the number of studies considered[61]. Review Manager (RevMan) v5.3 (Copenhagen: The Nordic Cochrane Center, The Cochrane Collaboration, 2014) and STATA 15 (StataCorp College Station, TX, USA) were used for statistical analyses. All tests were two sided, with $P < 0.05$ considered significant.

**Subgroup analysis and investigation of heterogeneity.** We conducted subgroup analysis based on whether the participants were influenza vaccine naive (clinical efficacy) or based on seroprotection rate after a single dose of non-adjuvanted vaccine (poor response: <25%; moderate response: 25 to 75%; and good response: >75%) (seroprotection). We assessed difference between subgroups using the test for heterogeneity between subgroups in RevMan 5.3. In the presence of relevant heterogeneity, we investigated the source of heterogeneity using random-effect meta-regression in STATA 15 (command: metareg). Meta-regression was performed using only one independent variable, the inverse of seroprotection rate in the non-adjuvanted arm, for which there existed a very large variation across studies (ranging from 4 to 100%).

**Risk of bias assessment.** Risk of bias was assessed using the Cochrane risk of bias tools. Risk of bias consisted of seven specific domains, including selection bias (random sequence generation and allocation concealment), performance bias (blinding of participants and personnel and other potential threats to validity), detection bias (blinding of outcome assessment and other potential threats to validity), attrition bias (incomplete outcome data), and reporting bias (selective outcome reporting assessed by comparing outcomes reported in the protocol to those reported in the completed RCTs whenever possible).

**Sensitivity analyses.** We performed sensitivity analysis by excluding studies with high risk of bias, as well as by excluding the trials that did not compare the same vaccine with or without adjuvants (using adjuvanted and non-adjuvanted vaccine from different manufacturers or using antigen-sparing design with reduced hemagglutinin antigen dose in adjuvanted vaccine).

**Grade of evidence.** We used the five GRADE considerations to assess the quality of evidence, that is, the risk of bias, inconsistency, indirectness, imprecision, and publication bias. We employed GRADEpro (https://gradepro.org/) to create summary tables of the findings for each outcome. We justified all decisions to downgrade or upgrade the quality of evidence using footnotes and comments.

**Reporting summary.** Further information on research design is available in the Nature Research Reporting Summary linked to this article.

## Data availability
All the data analyzed in this study are included in the published article. The source data underlying Figs. 2–10 and Supplementary Figs. 1–10 and 13–30 are provided as a Source Data file.

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

## Acknowledgements

This work is part of the Ph.D. dissertation of the first author Yu-Ju Lin, at National Taiwan University (2019). We thank the financial support provided by Taiwan Centers for Disease Control (Taipei, Taiwan). The funder has no role in the study design, data collection and analysis, or preparation of the manuscript.

## Author contributions

Y.-J.L. and C.-T.F. designed the study. Y.-J.L. and C.-N.W. independently applied the inclusion criteria to screen all identified and retrieved literature, and independently extracted the data. Y.-Y.L., W.-C.H. and C.-C.C. helped to identify and retrieve literature. Y.-J.L. conducted meta-analysis. Y.-H.C. conducted meta-regression. Y.-J.S. provided information on the evolution of CHMP regulatory guidelines. C.-T.F. and Y.-J.L. wrote the manuscript. Y.-H.C., C.-H.H. and C.-H.C. critically reviewed the draft. All authors approved the final version of the manuscript.

## Competing interests

The authors declare no competing interests.
