## [Peer Review File · Nature Communications]

Reviewers' Comments:

Reviewer #1:

Remarks to the Author:

I have comments relating to two linked problems throughout the review.

The aim of the review is not clearly stated. The following statement "Here, we conducted the first systematic review and meta-analysis of oil-in-water adjuvants for influenza vaccination in young children." Is ambiguous as it could be interpreted either to mean that the aim of the review assessed adjuvanted influenza vaccines or that it assessed adjuvants.

The title of the submission would seem to favour the latter interpretation: "Oil-in-water emulsion adjuvants for pediatric influenza vaccines: a systematic review and meta-analysis". However this ambiguity blights the rest of the review.

The meta-analyses mix, match and compare different vaccines, presumably in very different populations. This may very well explain the very high heterogeneity.

One further major problem is the complete absence of any mention of harms. Given the target population (children) and the subject matter (vaccines which have immunogenicity) this is unacceptable.

The searches are very basic and are inadequately described.

The Discussion and much of the details in Methods are missing.

There are factual mistakes in the text. For example:

adjuvants are not regulated as separate entities. The assertion from line 58 that MF59... was licensed..." is wrong. There is no licensing of adjuvants separately from the vaccines that contain them.

The statement "Inactivated influenza vaccines without adjuvant are poorly immunogenic in young children, who are particularly vulnerable to complications from influenza" is supported by reference to a Cochrane review which does not include serological outcomes not does it state that children are at highest risk of anything.

This is a clear misquote and another factual mistake.

I would suggest a complete revision before going any further to address these major problems. A clear aim would make everything clearer and easier.

Reviewer #2:

Remarks to the Author:

This systematic review on oil-in-water adjuvanted pediatric influenza vaccines is an important topic, well written and potentially useful for the field.

My major concerns relate to the systematic review methodology.

In particular I recommend that the authors ensure that they have reviewed the AMSTAR2 instrument to ensure that all the information required is included.

Screening and extraction should be done in duplicate, but this does not seem to have been the case. Supplementary information should include the characteristics of excluded studies.

Sources of funding for the included studies should be identified, as well as for this systematic review.

I am also concerned that the initial search results did not yield that many papers, and the search strategy may have been more specific than sensitive. Please could the authors comment on this. Also,

were wildcard terms included?

The stratification by level of response is an interesting way to address heterogeneity, but it is a little harder to see how this might be used to inform policy decisions when making decisions in advance of knowing which strains would be circulating and what level of response would be anticipated in any particular population or risk group. This should be discussed further as it speaks to the utility and potential impact of this review.

Minor comments:

Introduction: Note that LAIV was also recommended for children in Canada from the 2011/12 season.

The authors should discuss the influence of the nature of circulating influenza strains since this may influence relative and absolute vaccine efficacy seen in trials, as well as the power to detect differences.

Note that while RCTs are clearly important, field evaluation of influenza vaccines is equally so. The authors should acknowledge that the highly controlled conditions of an RCT need to be followed up with quantifying real-world experience of vaccine effectiveness.

Reviewer #3:

Remarks to the Author:

NCOMMS-18-29172

The paper has reviewed studies of adjuvanted influenza vaccines for children.

The figures are a little confusing. The column headings, for example, say "Risk ratio" but it isn't really clear what they are a ratio of. If seroprotection then that should appear somewhere in the headings. It also seems a little odd to refer to a ratio of sero-protection as a risk ratio, which would ordinarily be defined by a binary event. Moreover, the proportion sero-protected is referred to as a rate (which is isn't), leading to further confusion in the figures. Thus some clearer terminology is warranted.

It is unclear how heterogeneity can be reliably measured when only 2 studies are compared? Just because the meta-analysis programme provides an I^2 and Cochran's Q doesn't mean that it is a meaningful statistic. The authors should consider what these values mean when only 2 or 3 studies are considered and think about whether they should bother including these statistics.

When heterogeneity is so high, it is questionable whether estimates should be pooled at all. See papers by Greenland on this topic; e.g. PMID: 8030632; PMID: 12933636; PMID: 10472946. A more informative analysis would be to extend the meta-regression. The authors have done this to some extent by stratifying studies according to response. Nevertheless, heterogeneity remains high within these groups and further analyses may be warranted to identify the sources of heterogeneity. In addition, the authors should discuss the validity of their pooled estimates under high heterogeneity and whether they should inform public health decision making when the weight of evidence is so flaky.

The public health implications of the key findings should be discussed. For example, if the use of adjuvant vaccine is only really indicated for young children with no prior exposures to influenza or influenza vaccine, how should a public health programme use this information? Are there examples which set such a precedent? i.e. a different regimen for the previously unexposed? And how would this even be assessed? While parents may be adamant that their child has had flu, without a PCR

confirmation a severe illness might just as easily be caused by RSV. Given these hurdles there is probably no practical role for adjuvanted formulations in seasonal vaccination, and this may need more overt acknowledgement.

To Reviewer 1:

Thank you for your positive response to our work and the kind advice. We greatly appreciate your constructive comments that have helped us improve our paper. We have endeavored to incorporate the feedback and revised our manuscript accordingly. The itemized response is as follows:

C1. The aim of the review is not clearly stated. The following statement “Here, we conducted the first systematic review and meta-analysis of oil-in-water adjuvants for influenza vaccination in young children.” Is ambiguous as it could interpret either to mean that the aim of the review assessed adjuvanted influenza vaccines or that it assessed adjuvants. The title of the submission would seem to favour the latter interpretation: “Oil-in-water emulsion adjuvants for pediatric influenza vaccines: a systematic review and meta-analysis”. However, this ambiguity blights the rest of the review. I would suggest a complete revision before going any further to address these major problems. A clear aim would make everything clearer and easier.

R1. We clearly specified the aim in the revised introduction:

“We aimed to evaluate whether oil-in-water adjuvanted inactivated influenza vaccines provide protection superior to that by non-adjuvanted counterparts for children less than 9 years old, in terms of clinical efficacy against reverse transcription polymerase chain reaction (RT-PCR)-confirmed influenza (any strain), seroprotection rate (the proportion with a $\geq 1:40$ HI titer), and the need for a second dose.” (page 3, ll. 68-72)

C2. The meta-analyses mix, match and compare different vaccines, presumably in very different populations. This may very well explain the very high heterogeneity.

R2. To investigate the source of heterogeneity, we performed meta-regression, which shows that the between-trials heterogeneity was nearly all explained by difference in serological response rates in the non-adjuvanted arm (a surrogate indicator for whether the young children participants were immunologically naive to influenza or influenza vaccines), with R-square of 99.8% for A/H1N1, 99.3% for A/H3N2, and 99.6% for B (Fig. 4, Fig. 6, and Fig. 8). (page 5, ll. 121-125)

C3. One further major problem is the complete absence of any mention of harms. Given the target population (children) and the subject matter (vaccines which have immunogenicity) this is unacceptable.

- R3.** We included analysis on harm (adverse events), in term of local pain/tenderness and fever after the vaccination. (page 6, ll. 160-171)
- C4.** The searches are very basic and are inadequately described. The Discussion and much of the details in Methods are missing.
- R4.** We re-do a comprehensive search, in line with the current standard of the field. We described the details in Method. (pages 10-13; Appendix 1-8 in Supplemental Material)
- C5.** There are factual mistakes in the text. For example: adjuvants are not regulated as separate entities. The assertion from line 58 that MF59... was licensed...” is wrong. There is no licensing of adjuvants separately from the vaccines that contain them.
- R5.** We revised the description as below:

“MF59-adjuvanted seasonal influenza vaccines were first licensed for the elderly in 1997.” (page 2, ll. 61-62)

- C6.** The statement “Inactivated influenza vaccines without adjuvant are poorly immunogenic in young children, who are particularly vulnerable to complications from influenza” is supported by reference to a Cochrane review which does not include serological outcomes not does it state that children are at highest risk of anything. This is a clear misquote and another factual mistake.
- R6.** We now cited Neuzil, K. M. *et al.* Immunogenicity and reactogenicity of 1 versus 2 doses of trivalent inactivated influenza vaccine in vaccine-naive 5-8-year-old children. *J Infect Dis* **194**, 1032-1039 (2006) (reference 2), to support the statement: “inactivated influenza vaccines without adjuvant are poorly immunogenic in young children.”

We also cited the following three references (as reference 3-5) to support the statement “young children are at high risk for influenza-associated complications.”

Thompson, W.W. *et al.* Influenza-Associated Hospitalizations in the United States. *JAMA* **292**, 1333-40 (2004).

Neuzil KM1, Mellen BG, Wright PF, Mitchel EF Jr, Griffin MR. The effect of influenza on hospitalizations, outpatient visits, and courses of antibiotics in children. *N Engl J Med* **342**, 225-31 (2000).

Glaser CA *et al.* A Population-Based Study of Neurologic Manifestations of Severe Influenza A(H1N1)pdm09 in California. *Clin Infect Dis* **55**, 514-20 (2012).

To Reviewer 2:

Thank you for your positive response to our work and the kind advice. We greatly appreciate your constructive comments that have helped us improve our paper. We have endeavored to incorporate the feedback and revised our manuscript accordingly. The itemized response is as follows:

- C1.** This systematic review on oil-in-water adjuvanted pediatric influenza vaccines is an important topic, well written and potentially useful for the field. My major concerns relate to the systematic review methodology. In particular, I recommend that the authors ensure that they have reviewed the AMSTAR2 instrument to ensure that all the information required is included. Screening and extraction should be done in duplicate, but this does not seem to have been the case. Supplementary information should include the characteristics of excluded studies. Sources of funding for the included studies should be identified, as well as for this systematic review.

- R1.** We reviewed the AMSTAR2 instrument and updated our methodology to be in line with the current standard in the field. The original search was not done in duplicate. To meet the standard, two review authors re-do the search and extraction, using a comprehensive search strategy, in duplicate. We included the list of excluded studies and reasons for the exclusion in the supplementary information. Sources of funding for the included studies were also identified and listed in Table S1, S2, and S3. (pages 10-13, Appendix 1-8 and Tables S1-S3 in Supplemental Material)

- C2.** I am also concerned that the initial search results did not yield that many papers, and the search strategy may have been more specific than sensitive. Please could the authors comment on this. Also, were wildcard terms included?

- R2.** We re-do the search with a more sensitive strategy (Appendix 1-7 for Medline/PubMed, CENTRAL, Embase, CINAHL, google scholar, clinicaltrial.gov, and the grey literature, respectively), including the use of wildcard terms. (Supplemental Material)

- C3.** The stratification by level of response is an interesting way to address heterogeneity, but it is a little harder to see how this might be used to inform policy decisions when making decisions in advance of knowing which strains would be circulating and what level of response would be anticipated in any particular population or risk group. This should be discussed further as it speaks to the utility and potential impact of this review.

R3. “An age-based approach like that used for HPV vaccination would be desirable. This systematic review suggests that 3 years might be the best cut-off age, because the all-naive group (Fig. 2) and the poor response group (for whom adjuvanted vaccines are significantly superior to the non-adjuvanted counterparts) in Fig. 3, Fig. 5, and Fig. 7 are mostly comprised of young children aged less than 3 years while the non-naive group and the moderate/good response groups (for whom adjuvant and non-adjuvanted vaccines are similar) included many children aged 3 years or more. Therefore, oil-in-water adjuvanted influenza vaccine would be most cost-effective in the first seasonal influenza vaccination for young children less than 3 years old. On the other hand, for older children or children who had prior influenza vaccination, there might be no advantage to using oil-in-water-adjuvanted influenza vaccines to replace **less expensive** non-adjuvanted vaccines for annual influenza vaccination.” (page 9, ll. 253-264)

C4. Minor comments: Introduction: Note that LAIV was also recommended for children in Canada from the 2011/12 season.

R4. We added the information in Introduction:

“Live-attenuated influenza vaccines were also recommended for children in Canada from the 2011/2012 season ¹⁴.” (page 2, ll. 50-51)

C5. The authors should discuss the influence of the nature of circulating influenza strains since this may influence relative and absolute vaccine efficacy seen in trials, as well as the power to detect differences.

R5. We added these important points in Discussion:

“Mismatch between vaccine strains and the circulating influenza strains may substantially decrease both the absolute vaccine efficacy and the relative advantage of adjuvanted vaccines seen in trials, as well as the statistical power to detect differences.” (page 10, ll. 284-287)

C6. Note that while RCTs are clearly important, field evaluation of influenza vaccines is equally so. The authors should acknowledge that the highly controlled conditions of an RCT need to be followed up with quantifying real-world experience of vaccine effectiveness.

R6. We added these important points in Discussion:

“Pre-licensing RCTs need to be followed by post-marketing field evaluation to quantify the real-world experience of vaccine effectiveness. The present systematic review was essentially based on data from highly controlled conditions of RCTs. We were unable to find any field evaluation reports, likely due to the withdrawal of Fluvad Pediatric™ in 2012.” (page 10, ll. 287-291)

To Reviewer 3:

Thank you for your positive response to our work and the kind advice. We greatly appreciate your constructive comments that have helped us improve our paper. We have endeavored to incorporate the feedback and revised our manuscript accordingly. The itemized response is as follows:

C1. The figures are a little confusing. The column headings, for example, say “Risk ratio” but it isn’t really clear what they are a ratio of. If seroprotection then that should appear somewhere in the headings. It also seems a little odd to refer to a ratio of seroprotection as a risk ratio, which would ordinarily be defined by a binary event. Moreover, the proportion seroprotected is referred to as a rate (which is isn’t), leading to further confusion in the figures. Thus, some clearer terminology is warranted.

R1. We revised the column headings to “seroprotection rate ratio” in the forest plots showing analysis on seroprotection (Fig. 3, Fig. 5, Fig. 7). In the original clinical trial reports, the term “seroprotection rates” represents the proportions of subjects with titer reaching the seroprotection level. We specified the definition of seroprotection rate in page 3, ll. 71-71.

“seroprotection rate (the proportion with a $\geq 1:40$ HI titer)”

C2. It is unclear how heterogeneity can be reliably measured when only two studies are compared? Just because the meta-analysis programme provides an I^2 and Cochran’s Q doesn’t mean that it is a meaningful statistic. The authors should consider what these values mean when only 2 or 3 studies are considered and think about whether they should bother including these statistics.

R2. Unlike Cochran’s Q, I^2 statistic is independent on the number of studies considered (Higgins and Thompson, 2002). Therefore, we add the following explanation:

“ I^2 statistic, which describes the percentage of variation across studies that is due to heterogeneity rather than chance and is independent on the number of studies considered⁵⁰.” (page 13, ll. 397-399)

C3. When heterogeneity is so high, it is questionable whether estimates should be pooled at all. See papers by Greenland on this topic; e.g. PMID: 8030632; PMID: 12933636; PMID: 10472946. A more informative analysis would be to extend the meta-regression.

The authors have done this to some extent by stratifying studies according to response. Nevertheless, heterogeneity remains high within these groups and further analyses may be warranted to identify the sources of heterogeneity. In addition, the authors should discuss the validity of their pooled estimates under high heterogeneity and whether they should inform public health decision making when the weight of evidence is so flaky.

- R3.** To investigate the source of heterogeneity, we performed meta-regression, which shows that the between-trials heterogeneity was nearly all explained by difference in serological response rates in the non-adjuvanted arm (a surrogate indicator for whether the young children participants were immunologically naive to influenza or influenza vaccines), with R-square of 99.8% for A/H1N1, 99.3% for A/H3N2, and 99.6% for B (Fig. 4, Fig. 6, and Fig. 8). (page 5, ll. 121-125)

Subgroup analysis further showed that pooled estimates for the poor response subgroups (adjuvanted vaccines superior to the non-adjuvanted) and that for the good response subgroups (both are similar) are homogeneous within the subgroup ($I^2 = 0\%$). Since these subgroups should not be pooled, **we removed the pooled total from forest plots on clinical efficacy (Fig. 2) and seroprotection (Fig. 3 to Fig. 6).** Only estimates for subgroups, and the results of testing subgroup difference, are shown in the revised figures. We discuss the validity of pooled estimates in light of the meta-regression results, and the implication for public health decision-making. (page 8, ll. 231-246) (page 9, ll. 247-264)

- C4.** The public health implications of the key findings should be discussed. For example, if the use of adjuvant vaccine is only really indicated for young children with no prior exposures to influenza or influenza vaccine, how should a public health programme use this information? Are there examples which set such a precedent? i.e. a different regimen for the previously unexposed? And how would this even be assessed? While parents may be adamant that their child has had flu, without a PCR confirmation a severe illness might just as easily be caused by RSV. Given these hurdles there is probably no practical role for adjuvanted formulations in seasonal vaccination, and this may need more overt acknowledgement.

- R4.** We discussed the public health implications of the key findings as below

“Our findings support a role of oil-in-water-adjuvanted inactivated influenza vaccines for influenza-naive healthy young children, but not for children who had past influenza infections. A similar precedent is HPV vaccines, which are highly protective for

HPV-naïve girls but not protective for women who had already been infected with HPV of vaccine types ⁴⁶. To simplify the implementation of public health programme, HPV vaccines are now recommended for girls aged 11 years but not for women aged 26 years or more ⁴⁷. Because flu-like symptoms in young children may be caused by other virus such as RSV, an age-based approach like that used for HPV vaccination would be desirable. This systematic review suggests that 3 years might be the best cut-off age, because the all-naïve group (Fig. 2) and the poor response group (for whom adjuvanted vaccines are significantly superior to the non-adjuvanted counterparts) in Fig. 3, Fig. 5, and Fig. 7 are mostly comprised of young children aged less than 3 years while the non-naïve group and the moderate/good response groups (for whom adjuvant and non-adjuvanted vaccines are similar) included many children aged 3 years or more. Therefore, oil-in-water adjuvanted influenza vaccine would be most cost-effective in the first seasonal influenza vaccination for young children less than 3 years old. On the other hand, for older children or children who had prior influenza vaccination, there might be no advantage to using oil-in-water-adjuvanted influenza vaccines to replace **less expensive** non-adjuvanted vaccines for annual influenza vaccination.” (page 9, ll. 247-264)

Thank you!

Reviewers' Comments:

Reviewer #1:

Remarks to the Author:

The majority of the outcomes chosen by the authors are de facto surrogate (serological), because very few trials have clinical outcomes (3/18 according to the authors). The relevance of serological outcomes is highly dubious as the authors acknowledge in the text, especially in children. EMA regulators decided in 2014 to no longer use serological criteria (as cited in the manuscript). So why include such outcomes in the analysis?

Also note that the authors downgrade surrogate reporting because of bias (page 7).

The issue of analysing populations of children by their immune status or immunisation records is controversial as it does not relate to current policy and practise. These recommend vaccination of all children yearly and do not distinguish by immune status or immunisation history. This point should be added to the limitations of the evidence.

My comments about harms outcomes have been brushed aside. I am unclear whether the harm outcomes analysed by the authors were the only ones reported in the trials or were selected by the authors. There are also missing definitions of the outcomes (probably in the original trials). My point is the same. Use of adjuvanted vaccines in children is seen by some as controversial. Failure to report harms fuels conspiracy theories, so the authors should comment on the standard of reporting and make clear the rationale for their choice of harms outcomes. Also note that they seem to imply that a 12 month follow up for other outcomes translated in a 7 day follow up for harms (11). That needs checking or clarifying or both.

Another unclear point is whether the included trials tested the SAME vaccine from the SAME manufacturer with and without adjuvant. This is vital to introduce comparability in biologics.

The choice of forest plots is peculiar. If you look at Figures 2,3, 5,7, 9 and 10 (harms), adjuvanted vaccines fair better than their non adjuvanted counterparts ONLY in the naive subset of 4839 which is less than a third of the total of 15,310 participants in the trials with clinical outcomes. In all other forest plots the adjuvanted vaccines perform worse.

This does not translate into the conclusions from line 292: adjuvanted vaccines "have good efficacy" in naive children whereas "in primed children there may be no clear advantage".

The authors' bias is quite clear in this statement.

Reviewer #2:

Remarks to the Author:

Thank you to the authors who appear to have responded appropriately to all the issues raised.

Reviewer #4:

Remarks to the Author:

I was asked to comment as to whether the authors have satisfactorily responded to the comments from Reviewer 3. The responses and revisions are appropriate. I share the concern of reviewer 3 that the I^2 statistic is not meaningful with only 2 studies. The authors note that it is independent of the number of studies, and they elected to report it even when only 2 studies were analyzed. I don't think the I^2 adds any additional useful information for only 2 studies, but I don't have a strong objection

to reporting it. My only other comment pertains to the discussion of public health implications (R4). The phrase "...would be most cost-effective in..." should be replaced with "may be most appropriate for" since the authors did not conduct a cost-effectiveness analysis.

To Editor:

Thank you for your positive response to our work and the kind advice. We greatly appreciate your constructive comments that have helped us improve our paper. We have endeavored to incorporate the feedback and revised our manuscript accordingly. The itemized response is as follows:

C1. Given the remaining concerns on the use of serological markers as surrogate of protection, please do not make claims on 'clinical protection' (e.g. in the abstract), unless the claim is supported by trials that report clinical outcomes.

R1. We revised the abstract as below:

“Compared with non-adjuvanted counterparts, adjuvanted influenza vaccines provided a significantly better protection (weighted estimate for risk ratio of RT-PCR-confirmed influenza: 0.26) and were significantly more immunogenic (weighted estimates for seroprotection rate ratio: 4.6 to 7.9) in healthy immunologically naive young children.”
(pages 1-2, ll. 27-30)

Elsewhere in the manuscript, the use of “clinical protection” has already been supported by trials that reported clinical outcomes.

C2. Please also make sure to clarify the harms analyses

R2. We now presented serious adverse event (SAE) and neurological event during follow-up period (ranging from 21 days to 390 days) as the main harm outcomes. We provided definitions for these two harm outcomes of particular concern. The details of trials which reported these two harm outcomes were shown in new Supplementary Table 3 and Table 4. The meta-analysis results of SAE and neurological event were shown in new Figure 9 and Figure 10.

Harm outcomes (and their definitions) were clearly stated in the Introduction and Methods:

“We also assessed harm outcomes, including serious adverse event (SAE, defined as events which are life-threatening, resulting in death, required hospitalization, resulting in a residual disability, or required intervention to prevent permanent impairment or damage), neurological events (defined as events which involved central or peripheral

nervous system, including Guillain-Barré syndrome, cranial nerve palsy, narcolepsy, multiple sclerosis, and other neuroinflammatory disorders), and reactogenicity (fever or local pain/tenderness at injection sites).” (page 3, ll. 73-79)

“Definition of SAE: SAE is defined as any post-vaccination event which are life-threatening, resulting in death, required hospitalization, resulting in a residual disability, or required intervention to prevent permanent impairment or damage. SAE may or may not be vaccine related.” (page 13, ll. 389-392)

“Definition of neurological event: Neurological event was defined as any post-vaccination event involving central or peripheral nervous system, including Guillain-Barré syndrome, cranial nerve palsy, narcolepsy, multiple sclerosis, and other neuroinflammatory disorders.” (page 13, ll. 394-397)

We described the results of SAE analysis:

“Fifteen trials reported a total of 1,138 SAEs, which may or may not be considered vaccine-related by trial investigators, from 25,541 participants during follow-up period (ranging from 21 days to 390 days after vaccination, Supplementary Table S3). The most common SAE were gastroenteritis, pneumonia, convulsion, febrile convulsion, appendicitis, and asthma.” (page 6, ll. 172-175)

“Compared with participants in non-adjuvanted arms, those in adjuvanted arms did not have a higher frequency of SAE (weighted SAE risk ratio: 0.9, 95% CI: 0.7 to 1.1, I^2 : 50%, Fig. 9) during the follow-up period.” (page 6, ll. 176-178)

“The certainty of evidence was very low for SAE endpoint due to potential outcome reporting bias (only 15 of the 18 included trials reported SAE data), heterogeneity between trials (I^2 50%), and wide confidence interval.” (page 8, ll. 239-241)

We described the results of neurological event analysis:

“Fifteen trials reported a total of 13 neurological events (9 in adjuvanted arm and 4 in non-adjuvanted arm) from 20,802 participants during follow-up period (ranging from 21 days to 390 days after the vaccination, Supplementary Table S4). These neurological events include febrile convulsion (7/13), convulsion (3/13), and multiple seizure (3/13).” (page 6, ll. 181-184)

“Compared with participants in non-adjuvanted arms, those in adjuvanted arms had a slightly higher frequency of neurological events during the follow-up period, but the difference did not reach statistical significance due to the very low frequency of events (weighted risk ratio estimate: 1.4, 95% CI: 0.4 to 4.3, I^2 : 0%, Fig. 10).” (page 7, ll. 185-188)

“The certainty of evidence was low for neurological event endpoint due to potential outcome reporting bias (only 15 of the 18 included trials reported neurological events data), and few events with a wide confidence interval.” (page 8, ll. 242-244)

We also discussed our findings in the context of new data on Pandemrix and narcolepsy:

“We observed a slightly increase in frequency of neurological events among participants in adjuvanted vaccine arm, but the association is statistically inconclusive under the sample size of 20,802 participants in total from 15 RCTs. In 2010, an increase in narcolepsy cases among children/adolescents vaccinated with Pandemrix (AS03-adjuvanted H1N1 pdm09-monovalent vaccine made by GlaxoSmithKline) was detected in Europe, with an absolute risk of one additional case per 12,000 to 27,800 vaccinated. Antibodies to nucleoproteins of inactivated influenza virus in Pandemrix, which cross-react with human hypocretin receptor 2, may trigger the immune-mediated pathogenesis of narcolepsy. The other two AS03- or MF59-adjuvanted H1N1 vaccines, Arepanrix (GlaxoSmithKline) and Focetria (Novartis), contain a much lower amount of viral nucleoproteins than that in Pandemrix, and were not associated with a significant increase in risk of narcolepsy. When developing future oil-in-water-adjuvanted influenza vaccines, a better purification process during manufacturing to remove viral nucleoproteins is required to ensure vaccine safety.” (pages 11-12, ll. 335-347)

C3. Please clarify the types of vaccines and adjuvants used in the trials,

R3. We provided the details on the types of vaccines and adjuvants used in the trials in revised Supplementary Tables 2-5 in Supplementary Material. Six of the 18 trials compared the same vaccine with or without adjuvants. The other trials compared adjuvanted and non-adjuvanted vaccines from different manufacturers (8 trials) or used antigen-sparing design with a reduced antigen dose in adjuvanted vaccines arm (5 trials).

One of the rationales to use adjuvant is to reduce the hemagglutinin antigen dose (antigen-sparing). Efficacy data from trials using antigen-sparing design provided a more realistic estimate for adjuvanticity in future implementation scenarios. We

therefore still included these trials in the main analysis. To address the justified concern over comparability, we conducted a sensitivity analysis that included only trials which compared the same vaccine. Exclusion of the 12 trials which did not compare the same vaccine did not significantly alter the meta-analysis results.

We described these results in detail:

“Six of the 18 trials compared the same vaccine with or without adjuvants. The other trials compared adjuvanted and non-adjuvanted vaccines from different manufacturers (8 trials) or used antigen-sparing design with a reduced antigen dose in adjuvanted vaccines arm (5 trials).” (pages 4, ll. 84-97)

“Exclusion of the 12 trials which did not compare the same vaccine (with or without adjuvants) did not significantly alter the weighted estimate for risk ratio of RT-PCR-confirmed influenza in all naive group (0.2, 95% CI: 0.1 to 0.5), the weighted estimate for seroprotection rate ratio in the poor response group (H1N1: 4.6, 95% CI: 4.1 to 5.2; H3N2: 7.9, 95% CI: 6.7 to 9.4; B: 5.0, 95% CI: 4.4 to 5.7), the meta-regression results (R^2 99.56% for H1N1, 100% for H3N2, 99.84% for B), the weighted estimate for SAE risk ratio (0.7, 95% CI: 0.5 to 0.8), as well as the weighted estimate for risk ratio of neurological events (2.8, 95% CI: 0.1 to 67.5). (Fig. S19–S27).” (pages 7-8, ll. 215-222)

C4. Make textual changes to ensure that your conclusions are supported by the provided forest plots

R4. We apologize for mislabeling the rightward horizontal axis below the forest plots in Fig. 3, Fig. 5, and Fig. 7 as “Favor Control” (a higher seroprotection rate ratio should favor the experimental group, rather than the control group). The corrected Fig. 3, Fig. 5, and Fig. 7 (now labeling the rightward horizontal axis as “Favor Experimental”) show that the adjuvanted vaccines perform better than non-adjuvanted vaccine in naïve children. The revised Figures 2, 3, 5, 7 now support the conclusion that adjuvanted vaccines "have good efficacy" in naive children whereas "in primed children there may be no clear advantage" (page 12, ll. 356-362).

C5. Please address the remaining concern on reviewer #4.

R5. We revised the discussion of public health implications as suggested by reviewer #4:

“Therefore, oil-in-water adjuvanted influenza vaccine may be most appropriate for the

first seasonal influenza vaccination for young children less than 3 years old.” (page 10, ll. 306-308)

- C6.** Furthermore, please provide a PRISMA checklist together with the revised manuscript. Please highlight all changes in the manuscript text file.
- R6.** We provide a PRISMA checklist together with the revised manuscript. All changes in the manuscript text file were highlighted in red color.

We also updated the systematic review and meta-analysis up to 31 May 2019 (page 14).

Thank you!

To Reviewer 1:

Thank you for your positive response to our work and the kind advice. We greatly appreciate your constructive comments that have helped us improve our paper. We have endeavored to incorporate the feedback and revised our manuscript accordingly. The itemized response is as follows:

- C1.** The majority of the outcomes chosen by the authors are de facto surrogate (serological), because very few trials have clinical outcomes (3/18 according to the authors). The relevance of serological outcomes is highly dubious as the authors acknowledge in the text, especially in children. EMA regulators decided in 2014 to no longer use serological criteria (as cited in the manuscript). So why include such outcomes in the analysis? Also note that the authors downgrade surrogate reporting because of bias (page 7).
- R1.** We used RT-PCR-confirmed influenza as the main outcome. However, it is important to look for any inconsistency between the clinical outcome and the serological outcomes. If inconsistency exists, then the certainty of evidence must be downgraded (based on the Cochrane GRADE approach to grade the evidence in a systematic review. See Cochrane website: <https://training.cochrane.org/introduction-grade>). Therefore, it is necessary to include serological outcomes in the analysis.
- C2.** The issue of analysing populations of children by their immune status or immunisation records is controversial as it does not relate to current policy and practise. These recommend vaccination of all children yearly and do not distinguish by immune status or immunisation history. This point should be added to the limitations of the evidence.
- R2.** Sometimes, analysing populations of children by their immune status or immunisation records is necessary to inform policy and practice. This was recently exemplified by the case of DengvaxiaTM (Sanofi), a tetravalent dengue vaccine which protects against severe virologically confirmed dengue diseases in dengue-seropositive children but increases the risk of precisely the same outcome in dengue-seronegative children. We added this point in the discussion:

“The need to analyze population of children by their baseline serostatus is recently exemplified by the case of DengvaxiaTM (Sanofi), a tetravalent dengue vaccine which protects against severe virologically confirmed dengue diseases in dengue-seropositive children but increases the risk of precisely the same outcome in dengue-seronegative children.” (page 11, ll. 330-334)

C3. My comments about harms outcomes have been brushed aside. I am unclear whether the harm outcomes analysed by the authors were the only ones reported in the trials or were selected by the authors. There are also missing definitions of the outcomes (probably in the original trials). My point is the same. Use of adjuvanted vaccines in children is seen by some as controversial. Failure to report harms fuels conspiracy theories, so the authors should comment on the standard of reporting and make clear the rationale for their choice of harms outcomes. Also note that they seem to imply that a 12 month follow up for other outcomes translated in a 7 day follow up for harms (11). That needs checking or clarifying or both.

R3. Increased reactogenicity (fever or local pain/tenderness at injection sites) is the only harm that reaches statistical significance. To address the reviewer’s justified concern, we now presented serious adverse event (SAE) and neurological event during follow-up period (ranging from 21 days to 390 days) as the main harm outcomes. We provided definitions for these two harm outcomes of particular concern. The details of trials which reported these two harm outcomes were shown in new Supplementary Table 3 and Table 4. The meta-analysis results of SAE and neurological event were shown in new Figure 9 and Figure 10.

Harm outcomes (and their definitions) were clearly stated in the Introduction and Methods:

“We also assessed harm outcomes, including serious adverse event (SAE, defined as events which are life-threatening, resulting in death, required hospitalization, resulting in a residual disability, or required intervention to prevent permanent impairment or damage), neurological events (defined as events which involved central or peripheral nervous system, including Guillain-Barré syndrome, cranial nerve palsy, narcolepsy, multiple sclerosis, and other neuroinflammatory disorders), and reactogenicity (fever or local pain/tenderness at injection sites).” (page 3, ll. 73-79)

“**Definition of SAE:** SAE is defined as any post-vaccination event which are life-threatening, resulting in death, required hospitalization, resulting in a residual disability, or required intervention to prevent permanent impairment or damage. SAE may or may not be vaccine related.” (page 13, ll. 389-392)

“**Definition of neurological event:** Neurological event was defined as any post-vaccination event involving central or peripheral nervous system, including

Guillain-Barré syndrome, cranial nerve palsy, narcolepsy, multiple sclerosis, and other neuroinflammatory disorders.” (page 13, ll. 394-397)

We described the results of SAE analysis:

“Fifteen trials reported a total of 1,138 SAEs, which may or may not be considered vaccine-related by trial investigators, from 25,541 participants during follow-up period (ranging from 21 days to 390 days after vaccination, Supplementary Table S3). The most common SAE were gastroenteritis, pneumonia, convulsion, febrile convulsion, appendicitis, and asthma.” (page 6, ll. 172-175)

“Compared with participants in non-adjuvanted arms, those in adjuvanted arms did not have a higher frequency of SAE (weighted SAE risk ratio: 0.9, 95% CI: 0.7 to 1.1, I^2 : 50%, Fig. 9) during the follow-up period.” (page 6, ll. 176-178)

“The certainty of evidence was very low for SAE endpoint due to potential outcome reporting bias (only 15 of the 18 included trials reported SAE data), heterogeneity between trials (I^2 50%), and wide confidence interval.” (page 8, ll. 239-241)

We described the results of neurological event analysis:

“Fifteen trials reported a total of 13 neurological events (9 in adjuvanted arm and 4 in non-adjuvanted arm) from 20,802 participants during follow-up period (ranging from 21 days to 390 days after the vaccination, Supplementary Table S4). These neurological events include febrile convulsion (7/13), convulsion (3/13), and multiple seizure (3/13).” (page 6, ll. 181-184)

“Compared with participants in non-adjuvanted arms, those in adjuvanted arms had a slightly higher frequency of neurological events during the follow-up period, but the difference did not reach statistical significance due to the very low frequency of events (weighted risk ratio estimate: 1.4, 95% CI: 0.4 to 4.3, I^2 : 0%, Fig. 10).” (page 7, ll. 185-188)

“The certainty of evidence was low for neurological event endpoint due to potential outcome reporting bias (only 15 of the 18 included trials reported neurological events data), and few events with a wide confidence interval.” (page 8, ll. 242-244)

We also discussed our findings in the context of new data on Pandemrix and narcolepsy:

“We observed a slightly increase in frequency of neurological events among participants in adjuvanted vaccine arm, but the association is statistically inconclusive under the sample size of 20,802 participants in total from 15 RCTs. In 2010, an increase in narcolepsy cases among children/adolescents vaccinated with Pandemrix (AS03-adjuvanted H1N1 pdm09-monovalent vaccine made by GlaxoSmithKline) was detected in Europe, with an absolute risk of one additional case per 12,000 to 27,800 vaccinated. Antibodies to nucleoproteins of inactivated influenza virus in Pandemrix, which cross-react with human hypocretin receptor 2, may trigger the immune-mediated pathogenesis of narcolepsy. The other two AS03- or MF59-adjuvanted H1N1 vaccines, Arepanrix (GlaxoSmithKline) and Focetria (Novartis), contain a much lower amount of viral nucleoproteins than that in Pandemrix, and were not associated with a significant increase in risk of narcolepsy. When developing future oil-in-water-adjuvanted influenza vaccines, a better purification process during manufacturing to remove viral nucleoproteins is required to ensure vaccine safety.” (pages 11-12, ll. 335-347)

- C4.** Another unclear point is whether the included trials tested the SAME vaccine from the SAME manufacturer with and without adjuvant. This is vital to introduce comparability in biologics.
- R4.** We provided the details on the types of vaccines and adjuvants used in the trials in revised Supplementary Tables 2-5 in Supplementary Material. Six of the 18 trials compared the same vaccine with or without adjuvants. The other trials compared adjuvanted and non-adjuvanted vaccines from different manufacturers (8 trials) or used antigen-sparing design with a reduced antigen dose in adjuvanted vaccines arm (5 trials).

One of the rationales to use adjuvant is to reduce the hemagglutinin antigen dose (antigen-sparing). Efficacy data from trials using antigen-sparing design provided a more realistic estimate for adjuvanticity in future implementation scenarios. We therefore still included these trials in the main analysis. To address the justified concern over comparability, we conducted a sensitivity analysis that included only trials which compared the same vaccine. Exclusion of the 12 trials which did not compare the same vaccine did not significantly alter the meta-analysis results.

We described these results in details:

“Six of the 18 trials compared the same vaccine with or without adjuvants. The other trials compared adjuvanted and non-adjuvanted vaccines from different manufacturers (8

trials) or used antigen-sparing design with a reduced antigen dose in adjuvanted vaccines arm (5 trials).” (pages 4, ll. 84-97)

“Exclusion of the 12 trials which did not compare the same vaccine (with or without adjuvants) did not significantly alter the weighted estimate for risk ratio of RT-PCR-confirmed influenza in all naive group (0.2, 95% CI: 0.1 to 0.5), the weighted estimate for seroprotection rate ratio in the poor response group (H1N1: 4.6, 95% CI: 4.1 to 5.2; H3N2: 7.9, 95% CI: 6.7 to 9.4; B: 5.0, 95% CI: 4.4 to 5.7), the meta-regression results (R^2 99.56% for H1N1, 100% for H3N2, 99.84% for B), the weighted estimate for SAE risk ratio (0.7, 95% CI: 0.5 to 0.8), as well as the weighted estimate for risk ratio of neurological events (2.8, 95% CI: 0.1 to 67.5). (Fig. S19–S27).” (pages 7-8, ll. 215-222)

- C5.** The choice of forest plots is peculiar. If you look at Figures 2, 3, 5, 7, 9 and 10 (harms), adjuvanted vaccines fair better than their non adjuvanted counterparts ONLY in the naive subset of 4839 which is less than a third of the total of 15,310 participants in the trials with clinical outcomes. In all other forest plots the adjuvanted vaccines perform worse. This does not translate into the conclusions from line 292: adjuvanted vaccines "have good efficacy" in naive children whereas "in primed children there may be no clear advantage". The authors' bias is quite clear in this statement.
- R5.** We apologize for mislabeling the rightward horizontal axis below the forest plots in Fig. 3, Fig. 5, and Fig. 7 as “Favor Control” (a higher seroprotection rate ratio should favor the experimental group, rather than the control group). The corrected Fig. 3, Fig. 5, and Fig. 7 (now labeling the rightward horizontal axis as “Favor Experimental”) show that the adjuvanted vaccines perform better than non-adjuvanted vaccine in naïve children. The revised Figures 2, 3, 5, 7 now support the conclusion that adjuvanted vaccines "have good efficacy" in naive children whereas "in primed children there may be no clear advantage" (page 12, ll. 356-362).

Thank you!

To Reviewer 4:

Thank you for your positive response to our work and the kind advice. We greatly appreciate your constructive comments that have helped us improve our paper. We have endeavored to incorporate the feedback and revised our manuscript accordingly. The itemized response is as follows:

C1. I was asked to comment as to whether the authors have satisfactorily responded to the comments from Reviewer 3. The responses and revisions are appropriate. I share the concern of reviewer 3 that the I^2 statistic is not meaningful with only 2 studies. The authors note that it is independent of the number of studies, and they elected to report it even when only 2 studies were analyzed. I don't think the I^2 adds any additional useful information for only 2 studies, but I don't have a strong objection to reporting it. My only other comment pertains to the discussion of public health implications (R4). The phrase "...would be most cost-effective in..." should be replaced with "may be most appropriate for" since the authors did not conduct a cost-effectiveness analysis.

R1. We revised the discussion of public health implications as below:

“Therefore, oil-in-water adjuvanted influenza vaccine **may** be most **appropriate for** the first seasonal influenza vaccination for young children less than 3 years old.” (page 10, ll. 306-308)

Thank you!

Reviewers' Comments:

Reviewer #2:

Remarks to the Author:

Considerable improvements have been made in the paper in response to previous reviewers' comments.

Major comments

1. Given that narcolepsy is a relatively rare adverse event following immunization and also the fact that there are frequently delays making the diagnosis, the language around safety of vaccination needs to be toned down since this outcome could not realistically be examined in the included studies either because they are too small (i.e. insufficient power), have too short follow up or both. The finding of higher frequency of neurological events is therefore concerning regardless of their statistical significance (lines 186-9). This should be flagged as a potential hazard that would require close and specialized monitoring. The first paragraph of the discussion seems to downplay this limitation in only mentioning local reactions. The discussion should also emphasize the need for very good vaccine safety systems to track such outcomes for adjuvanted paediatric vaccines.

2. Are the differences clinically meaningful? VE is a funny measure that is superficially intuitive but is in reality a relative risk/odds ratio that we talk about as if it were an absolute risk reduction.

3. Correlation coefficients. The near perfect correlation looks like it is driven strongly by a few outlying data points. I suggest repeating a sensitivity analysis with and without those outliers to see how the data points that cluster together further down the scale are as well correlated without the outliers.

4. Discussion line 300. HPV vaccination is not a good example of an age-specific recommendation because the reason for giving the vaccine early is to prevent infection, not to simplify the program. Evidence does not support HPV vaccine being effective once chronic HPV infection is established. Influenza changes every year, and does not cause chronic infections. The reasons for non-response in previously infected are very different.

5. Discussion line 334. Denguevaxia is another problematic example. It certainly illustrates the idea of increased risk in dengue-positive, but it doesn't illustrate the idea of lower efficacy. There are better examples from influenza such as: <https://www.ncbi.nlm.nih.gov/pubmed/21857263>

6. Need to check that axes are labelled correctly throughout including supplementary materials. Also need to add definitions of "poor", "moderate" and "good" response for the forest plots.

Minor comments

7. Introduction line 43: The recommendations given for children under 9 years of age are from US, and recommendations vary by jurisdiction, age and risk group. Recommendations vary by country. For example, UK is using LAIV. Probably best to specify the country source of that recommendation.

8. Discussion line 270 – need to explain what is meant by "validity of the RT-PCR method" and further information about the withdrawal.

9. Line 287 – suggest replacing "value" with "utility" because the former is slightly ambiguous in the context.

10. Recommendation for 3 years as cut-off age. This is a programmatic recommendation. I suggest more caution in framing it because many other considerations determine age-specific recommendations

11. Line 341 Note that narcolepsy following adjuvanted influenza vaccine was not just an issue in Europe.

12. Line 347 – suggest replacing "significant" with "detectable" risk of narcolepsy. A better purification process may reduce risk but it may not ensure vaccine safety. Long term safety of these adjuvanted vaccines is unknown.

To Reviewer 2:

Thank you for your positive response to our work and the kind advice. We greatly appreciate your constructive comments that have helped us improve our paper. We have endeavored to incorporate the feedback and revised our manuscript accordingly. The itemized response is as follows:

C1. Given that narcolepsy is a relatively rare adverse event following immunization and also the fact that there are frequently delays making the diagnosis, the language around safety of vaccination needs to be toned down since this outcome could not realistically be examined in the included studies either because they are too small (i.e. insufficient power), have too short follow up or both. The finding of higher frequency of neurological events is therefore concerning regardless of their statistical significance (lines 186-9). This should be flagged as a potential hazard that would require close and specialized monitoring. The first paragraph of the discussion seems to downplay this limitation in only mentioning local reactions. The discussion should also emphasize the need for very good vaccine safety systems to track such outcomes for adjuvanted paediatric vaccines.

R1. We agree. We flagged the higher frequency of neurological events as a potential hazard in the first paragraph of the discussion:

“Participants in adjuvanted vaccine arms had a small increase in frequency of neuro-logical events, which should be considered as a potential hazard that would require close and specialized monitoring regardless of statistical significance.” (page 9, ll. 270-273)

We also added a new paragraph in the discussion to emphasize the need for very good vaccine safety systems to track such outcomes for adjuvanted paediatric vaccines:

“No narcolepsy case was reported in the RCTs which were included in this systematic review and meta-analysis. The 13 reported neurological events were febrile convulsion or seizure. Given that narcolepsy is a rare adverse event following vaccination and the fact that there are frequently delays in making the diagnosis, this outcome could not be realistically examined in included trials because of the limited sample size (20,802 participants in total) and the relatively short follow-up duration (from 21 to 390 days). Therefore, the finding of an increased frequency of neurological events is concerning. This highlights the need for a very good vaccine safety system to tracking such

outcome for adjuvanted pediatric vaccines in future clinical trials.” (page 13, ll. 374-382)

C2. Are the differences clinically meaningful? VE is a funny measure that is superficially intuitive but is in reality a relative risk/odds ratio that we talk about as if it were an absolute risk reduction.

R2. Vesikari (2011) reported a clinically meaningful absolute risk reduction: adjuvanted vaccines had an absolute VE (absolute risk reduction, using a placebo arm as reference) of 86%, while non-adjuvant vaccines had an absolute VE of only 43%. The other trial in the all-naïve group, Nolan (2014), did not include an all-placebo arm, and therefore, was unable to report absolute VE:

“Vesikari (2011)²⁹, which included a placebo arm, reported an absolute vaccine efficacy (against RT-PCR-confirmed influenza) of 86% (95% CI: 74% to 93%) for adjuvanted vaccines and 43% (95% CI: 15% to 61%) for non-adjuvant counterparts. Nolan (2014)³⁴ did not include an all-placebo arm, and therefore, did not report absolute vaccine efficacy.” (page 4, ll. 114-118)

C3. Correlation coefficients. The near perfect correlation looks like it is driven strongly by a few outlying data points. I suggest repeating a sensitivity analysis with and without those outliers to see how the data points that cluster together further down the scale are as well correlated without the outliers.

R3. A sensitivity analysis without those outliers still show a near perfect correlation (R^2 99.67% for A/H1N1, and 96.21% for B) except for A/H3N2 (R^2 42.98%), in which the meta-regression became imprecise ($P=0.075$) when the number of data points dropped to only seven (Supplementary Figures S28-S30):

“To examine whether the near perfection correlation in meta-regression is driven by a few outlying data points, we conducted a sensitivity analysis without these potential outliers. After excluding those data points with an inverse of seroprotection rate in non-adjuvanted arm larger than 5, the R^2 of meta-regression model remains very high (R^2 99.67% for A/H1N1, and 96.21% for B) except that for A/H3N2 (R^2 42.98%), in which the meta-regression results became imprecise (slope = 0.47, $P=0.075$) when the number of data points decreased to only seven (Fig. S28-S30).” (page 8, ll. 230-236)

C4. Discussion line 300. HPV vaccination is not a good example of an age-specific

recommendation because the reason for giving the vaccine early is to prevent infection, not to simplify the program. Evidence does not support HPV vaccine being effective once chronic HPV infection is established. Influenza changes every year, and does not cause chronic infections. The reasons for non-response in previously infected are very different.

- R4.** We agree. We deleted this problematic example.
- C5.** Discussion line 334. Denguevaxia is another problematic example. It certainly illustrates the idea of increased risk in dengue-positive, but it doesn't illustrate the idea of lower efficacy. There are better examples from influenza such as: <https://www.ncbi.nlm.nih.gov/pubmed/21857263>
- R5.** We agree. We deleted this problematic example. We now cited Skowronski DM *et al. Pediatr Infect Dis J.* **30**, 833-839 (2011) as Reference 47 to support that serostatus of participants needs to be taken into account when designing future randomized trials for these products. (page 12, line 349)
- C6.** Need to check that axes are labelled correctly throughout including supplementary materials. Also need to add definitions of “poor”, “moderate” and “good” response for the forest plots.
- R6.** Understood and done.
- C7.** Introduction line 43: The recommendations given for children under 9 years of age are from US, and recommendations vary by jurisdiction, age and risk group. Recommendations vary by country. For example, UK is using LAIV. Probably best to specify the country source of that recommendation.
- R7.** We now specified the country source of that recommendation:
- “To improve the efficacy, a two-dose schedule (separated by a greater than 4-week interval) is currently recommended for children younger than 9 years in the United States” (page 2, ll. 41-43)
- C8.** Discussion line 270 – need to explain what is meant by “validity of the RT-PCR method” and further information about the withdrawal.

R8. We explain the validity problem and the subsequent withdrawal with more details:

“However, Good Clinical Practice (GCP) inspection found that “the kit for the extraction of viral nucleic acid in swab samples was not validated”, and that the RT-PCR technique used to detect influenza virus “did not incorporate an internal control”⁴⁴. The inspection also identified “critical issues affecting the reliability of recorded adverse event and suspected influenza cases”⁴⁵. The authors “re-tested all available original samples (1208 of 1216 samples) that were collected during the trial in a qualified laboratory using a validated assay”⁴⁵. Although “analyses of the re-test results were remarkably similar to the original analyses, confirming the original overall study conclusions regarding the high absolute and relative efficacy”⁴⁴, this MF59-adjuvanted paediatric influenza vaccine (Fluad PediatricTM) was subsequently withdrawn from Europe in 2012 due to non-compliance with GCP during this clinical trial⁴⁴.” (page 10, ll. 286-296)

C9. Line 287 – suggest replacing “value” with “utility” because the former is slightly ambiguous in the context.

R9. We agree that the word “value” is slightly ambiguous in the context. However, the word “utility” did not convey the exact meaning we intended to express. We therefore revise the wording from “value” to “numerical value”. (page 11, ll. 310-313)

C10. Recommendation for 3 years as cut-off age. This is a programmatic recommendation. I suggest more caution in framing it because many other considerations determine age-specific recommendations.

R10. We agree. We delete the claim on using 3 year as a “cut-off age”. We now framing the issue in a more cautious way:

“The all-naive group and the poor response group, for whom adjuvanted vaccines are significantly superior to the non-adjuvanted counterparts, are mostly comprised of young children aged less than 3 years while the non-naive group and the moderate/good response groups (for whom adjuvant and non-adjuvanted vaccines are similar) included many children aged 3 years or more. Therefore, oil-in-water adjuvanted influenza vaccine may be most appropriate for the first seasonal influenza vaccination for young children less than 3 years old.” (page 11, ll. 321-327)

C11. Line 341 Note that narcolepsy following adjuvanted influenza vaccine was not just an

issue in Europe.

R11. We agree. We summarized the investigations outside Europe as below:

“Narcolepsy incidences following other adjuvanted H1N1pdm09-monovalent vaccines were also investigated in Canada (where a different AS03-adjuvanted vaccine, Arepanrix™ from GlaxoSmithKline, was used)^{51,52}, South Korea (where MF59-adjuvanted vaccine Greenflu-S plus™ was used; the MF59 was from Novartis)⁵³, and Taiwan (where MF59-adjuvanted vaccine Focetria™ from Novartis was used)⁵⁴. Most of these investigations did not find an increase in narcolepsy incidence, except for that in Quebec, Canada, where a possible small increase (1 per million vaccine doses) was reported (although “a confounding effect of influenza infection cannot be ruled out”⁵²). In Taiwan, a three-fold increase in narcolepsy incidence occurred before the start of H1N1 vaccination; and influenza-like illness during 2009 H1N1 pandemic period was a strong risk factor (case-control matched odds ratio: 2.72, P = 0.0137) for the onset of narcolepsy in children⁵⁴. In China (where no adjuvanted H1N1pdm09-monovalent vaccines were used), narcolepsy incidence increased by 3-fold following the 2009 H1N1 pandemic but the majority (>90%) of Chinese narcolepsy cases had no influenza vaccination history⁵⁵.” (page 12, ll. 353-366)

C12. Line 347 – suggest replacing “significant” with “detectable” risk of narcoplepsy. A better purification process may reduce risk but it may not ensure vaccine safety. Long term safety of these adjuvanted vaccines is unknown.

R12. We agree. We made the following revisions:

“The other two AS03- or MF59-adjuvanted H1N1 vaccines, Arepanrix™ and Focetria™, contain a much lower amount of viral nucleoproteins than that in Pandemrix^{56,57}, and were not associated with a detectable increase in risk of narcolepsy^{48-50, 56-59}. However, it remains unknown whether removal of viral nucleoprotein from adjuvanted influenza vaccines, through a better purification process during manufacturing, would eliminate the risk of narcolepsy.” (page 12, ll.368-373)

“Another important limitation is the limited length of follow-up (only up to 390 days after vaccination). Thus, long-term safety of these adjuvanted vaccines remains unknown.” (page 13, ll.390-392)

Thank you!

Reviewers' Comments:

Reviewer #2:

None